**METHOD**

# DCATS: differential composition analysis for flexible single-cell experimental designs

Xinyi Lin[1,2], Chuen Chau[1], Kun Ma[1,2], Yuanhua Huang[1,3*] and Joshua W. K. Ho[1,2*]

*Correspondence:
yuanhua@hku.hk; jwkho@hku.hk

[1] School of Biomedical Sciences,
Li Ka Shing Faculty of Medicine,
The University of Hong Kong,
Pokfulam, Hong Kong SAR, China
[2] Laboratory of Data Discovery
for Health Limited (D24H), Hong
Kong Science Park, Hong Kong
SAR, China
[3] Department of Statistics
and Actuarial Science, Faculty
of Science, The University
of Hong Kong, Pokfulam, Hong
Kong SAR, China

## Abstract

Differential composition analysis — the identification of cell types that have statistically significant changes in abundance between multiple experimental conditions — is one of the most common tasks in single cell omic data analysis. However, it remains challenging to perform differential composition analysis in the presence of flexible experimental designs and uncertainty in cell type assignment. Here, we introduce a statistical model and an open source R package, DCATS, for differential composition analysis based on a beta-binomial regression framework that addresses these challenges. Our empirical evaluation shows that DCATS consistently maintains high sensitivity and specificity compared to state-of-the-art methods.

## Background

Single-cell RNA sequencing (scRNA-seq) is a high-throughput sequencing technology that enables researchers to probe transcriptomes of a large number of individual cells, and allows them to characterize different cell types in a heterogeneous population. It plays a critical role in strengthening our understanding of various biological systems, including embryogenesis, the development of different diseases, and how cells react to environmental stimuli [1, 2]. Recently, highly multiplexed strategies have been introduced to mix samples from different donors, conditions, or treatments with external molecular barcodes or intrinsic genetic makeups to achieve higher efficiency and lower batch effects [3–5]. In such multi-sample designs, analysis of the differential composition of cell types between two conditions is routinely applied.

From a scRNA-seq experiment, we can obtain the cell counts $N = \{n_1, ..., n_K\}$ of $K$ cell types by performing cell clustering, for example, with Louvain (a graph-based method) [6] or K-means, usually on reduced dimensions through a principal component analysis. Then, the obtained cell count vector $N$ is conventionally used to estimate the cell type composition abundance $\mu$, with the assumption that the clustering is unbiased and accurate. Thus, a few statistical methods can be directly applied to detect the cell types with differential composition abundance between conditions. Some statistical tools are

also developed specialized for scRNA-seq data. scCODA [7] assumes cell counts of different cell types follow a hierarchical Dirichlet-Multinomial distribution, which allows scCODA to model all cell types together. MiloR [8] evaluates differential abundance on smaller clusters in KNN graph, which are called "neighborhoods." It assumes that cell counts follow a negative binomial distribution and uses representative cells instead of all cells to improve program efficiency. DAseq [9] also makes use of KNN graph and calculates multiscale differential abundance scores by counting the numbers of cells coming from different biological states while varying k. This multiscale differential abundance score is what DAseq uses to infer cell states that have differential abundance. propeller [10] uses an empirical Bayes framework to enable information sharing between different samples and chooses different statistical methods (t-test or ANOVA) based on the number of conditions. Other tools are developed to handle different challenges for different data types. For instance, ANCOM-BC [11] uses a linear model with sample-specific offset terms to handle the estimation of different sampling fractions when analyzing microbiome data. diffcyt [12] uses functions from the R package edgeR [13] to test differential abundance of cell types. It uses an overdispersed Poisson model with empirical Bayes methods for information sharing between different cell types. However, its implementation was specifically designed for flow cytometry data, not single cell RNA-seq data.

Nevertheless, performing differential abundance analysis on single cell RNA-seq data with replicates remains a statistical challenge for multiple reasons. First, multi-level variability exists due to technical and biological reasons, such as a low number of biological replicates or a low number of cells for minor cell types. Recently, scDC [14], a Poisson regression-based method, has been introduced to account for the uncertainty not only between replicates but also clustering by leveraging bootstrap re-sampling, preferred with re-clustering. However, the Poisson noise model is not able to capture the over dispersion. Second, misclassification during the cell clustering step may introduce both systematic bias and uncertainty. For example, subtypes of T helper cells are often confused with each other. A similar challenge was also noticed in meta-genomics analysis, where species with similar sequences are often confusedly aligned and quantified, and bias correction by reversing this bias was found beneficial for the differential abundance analysis [15, 16]. Regarding to challenges mentioned above, we introduced a statistical method, DCATS, to effectively detect the cell types with differential abundance between conditions or along with continuous covariates. This algorithm is implemented as an R package named DCATS which is available at https://github.com/holab-hku/DCATS.

## Results

### High-level description of DCATS

Here, we introduce DCATS, a differential composition analysis framework to address the above challenges. DCATS has two key features. First, it accounts for the uncertainty in cell type assignment utilizing the biased similarity between cells types. The latent true cell type proportion is obtained by leveraging a similarity matrix between cell types via maximum likelihood estimation. Second, DCATS employs a beta-binomial regression model to analyze the differential cell type abundance, which models the raw cell counts rather than the normalized proportions and considers the dispersion between samples (Fig. 1A; "Method"). Specifically, we assume the corrected cell counts $z_{s,j}$ for cell type $j$ given sample $s$ follow a

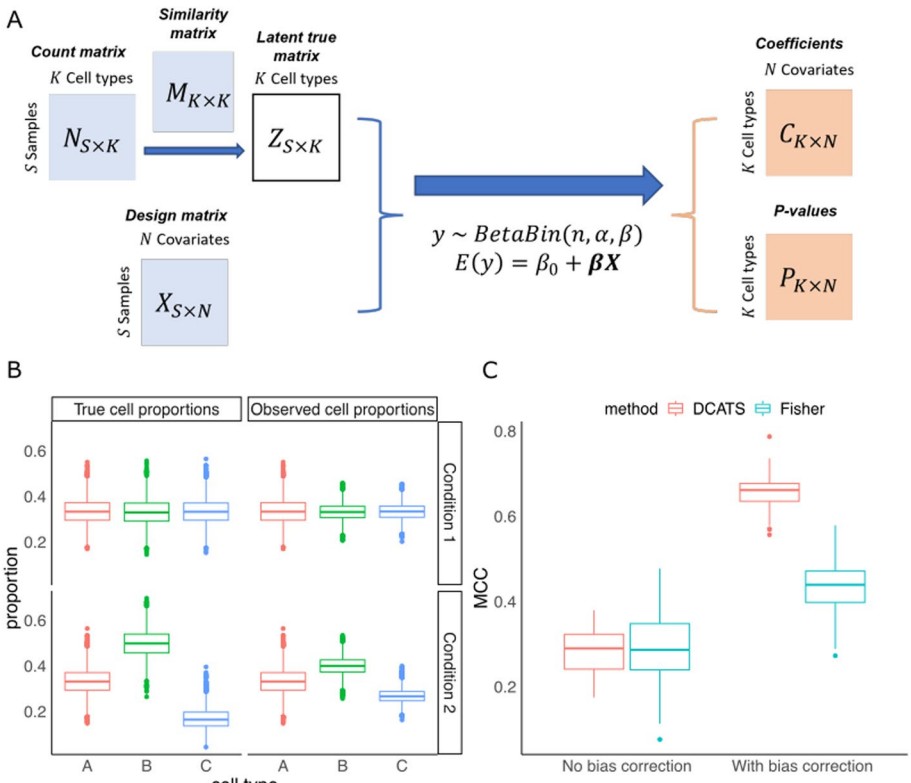

**Fig. 1** DCATS improves composition analysis through accounting for uncertainty in classification of cell types in differential abundance analysis. **A** Illustration of the DCATS workflow. Matrices with light blue are input matrices, matrices with light orange are output from DCATS. First step is bias correction using a similarity matrix. This step is optional. The design matrix with multiple covariates is also required. DCATS supports both categorical and continuous covariates. Then, DCATS detects differential abundance using a beta-binomial generalized linear model (GLM) model, which returns the estimated coefficients and *p*-values. **B** These box plots illustrate the effect of cell type misclassificaiton in a theoretical simulation with Dirichlet-Multinomial sampling. The similarity matrix is designed to introduce misclassification errors. The proportions of cell types **B** and **C** are changed between conditions 1 and 2. **C** Area under the precision-recall curve (AUC, same below) values when varying *p* value in detecting the cell type with differential abundance

beta-binomial distribution. We then describe $z_{s,j}$ with a beta-binomial generalized linear model(GLM) with a logit link function for each covariate *i*, as follows,

$$
\begin{aligned}
z_{s,j} &\sim \texttt{Binom}(n_s, p_{s,j}) \\
p_{s,j} &\sim \texttt{Beta}(\alpha_{s,j}, \beta_{s,j})
\end{aligned}
\tag{1}
$$

where we reparameterized with the mean $\bar{p}_{s,j}$ and over dispersion term $\phi_j$ as

$$
\begin{aligned}
\alpha_{s,j} &= \bar{p}_{s,j}(1/\phi_j - 1) \\
\beta_{s,j} &= (1 - \bar{p}_{s,j})(1/\phi_j - 1).
\end{aligned}
\tag{2}
$$

As described below, the mean parameter $\bar{p}_{s,j}$ is regressed to a set of covariates under different hypotheses and can be parameterized as

$$
\texttt{logit}(\bar{p}_{s,j}) = w_0 + \boldsymbol{w}^\top \boldsymbol{c}_s
\tag{3}
$$

where $c_s$ is the covariate vector for sample $s$, and $w$ is the corresponding weight vector. The weights for covariates $w$ and the dispersion $\phi_j$ are optimized to achieve maximum likelihood, through the *aod* package [17]. The $p$-value can be calculated with a likelihood ratio test by comparing the log likelihoods in both alternative and null hypotheses.

The effectiveness of the vanilla beta-binomial regression was demonstrated in [18]. However, the estimation of the over dispersion term in this task is often challenging due to the extremely low numbers of replicates (sometimes even single replicate). Therefore, we introduce a strategy to estimate the dispersion term for all cell types jointly, through a pooled beta-binomial regression in a pre-step (Method).

When analyzing scRNA-seq data, we started from the gene expression matrix of read or unique molecular identifier (UMI) counts. After basic pre-processing including filtering cells, normalization, and integration [19, 20], several methods can be used to annotate each cells, including manual annotation, supervised methods and semi-supervised methods [21, 22]. The input cell count matrix for DCATS can be calculated by counting the number of cells in each cell type.

In order to estimate the cell type similarity matrix $M$ for the confusion during clustering, we introduced two heuristic methods here. One is based on the KNN graph between cells by calculating the averaged frequency of cell types in each cell's neighbors (Method), where the KNN graph may be pre-computed if using Seurat [23] or Scanpy [24] pipelines. By defining $n_{x,j}$ as the number of neighbors for cell $x$ that are classified as cell type $j$, we can calculate the KNN-based similarity matrix, for example, its entry $m_{i,j}$ for similarity (misclassification) from cluster $i$ to $j$ as follows:

$$m_{i,j} = \frac{\sum_{x=1}^{x_i} n_{x,j}}{\sum_{j=1}^{K} \sum_{x=1}^{x_i} n_{x,j}}$$

The other method is to use a prediction confusion matrix produced using 5-fold cross validation by a classifier on top of principle components when the KNN graph is not available, e.g., support vector machine as default.

### Correcting misclustering through DCATS improves composition analysis

To evaluate the effects of misclustering on the composition analysis, we performed a theoretical simulation by generating cell counts from Dirichlet-Multinomial distributions, where the input cell type proportions were generated by adding bias to the genuine cell type proportions through a transformation with the misclustering matrix, aka similarity matrix (Fig. 1C, Additional file 1: Fig. S1).

Here, the genuine proportions of the three cell types were [1/3, 1/3, 1/3] and [1/3, 1/2, 1/6] in conditions 1 and 2, respectively. However, due to the high similarity between cell types 2 and 3, the average cell type proportions were transformed to [0.33, 0.33, 0.33] and [0.33, 0.40, 0.27] respectively in the two conditions, which mimicked the bias introduced in the clustering. Due to the equal cell type proportion and symmetrical similarity matrix, all cell type proportions remained unchanged in condition 1 after the similarity-based transformation. However, the proportion of cell type 2 and cell type 3 in condition 2 changed due to misclassification. Notably, the difference in proportions between conditions was reduced for both cell types 2 and 3, which caused false negatives due to the

shrunken effects. Indeed, both Fisher's exact test and DCATS only returned moderate performance (area under ROC curve (AUC) = 0.721 and 0.745 on average, aggregating 50 runs). By contrast, we found that the performance in detecting differential abundance was dramatically increased for both DCATS (mean AUC: 0.745 to 0.976) and Fisher's exact test (mean AUC: 0.721 to 0.961; Fig. 1C) by using the cell counts corrected from similarity matrix by DCATS through an Expectation-Maximization algorithm. This was largely thanks to the improved sensitivity in both methods (Additional file 1: Fig. S2).

We further asked whether the bias correction component in DCATS could correct the misclassification and provide more accurate proportion estimations. We tested this using a single cell RNA-seq data set [25] consisting of progressive multiple sclerosis (MS) and relapsing-remitting disease course MS (RRMS). In this dataset, cells from 71 PBMC samples of 62 donors were collected for single-cell RNA sequencing and surface antibody staining. In total, it contains 497,705 single-cell transcriptomic (Tr) and 355,433 surface protein (SP) profiles. The original study used defined cell types based on both Tr and SP profiles [25]. We treated this Tr+SP annotation as the "original" annotation with high reliability, as it is the most well-studied annotation [25].

To test bias correction in our study, we focused on four T-cell subtypes based on the Tr+SP annotated dataset, namely T06, T07, T09, and T10. Then, we identified cell clusters based on transcriptomic data only (Tr-only) and match each Tr-only cluster to its closest Tr+SP cluster. A confusion matrix can be generated between the four clusters defined by Tr+SP and their matching Tr-only clusters. The proportions of the four Tr-only clusters can be calculated. Using this "empirical" confusion matrix for bias correction, DCATS corrected the proportions of the four Tr-only clusters very close to the true proportions (root mean square error (RMSE) from 0.1810 to $8.672 \times 10^{-5}$, Additional file 1: Fig. S3). In most cases, the true similarity matrix or confusion matrix is unknown. DCATS provides two simple methods to approximate this similarity matrix, which improves our estimation of proportion by 44% compared to without bias correction (RMSE from 0.1810 to 0.1023, using KNN similarity matrix; Additional file 1: Fig. S3)

Combined with previous simulation results, our findings support the effectiveness of DCATS' bias correction step in differential composition analysis.

### Benchmarking DCATS with simulated data

We further benchmarked DCATS's performance with six existing methods: Fisher's exact test, scDC [14], speckle [10], diffcyt (a method primarily designed for mass cytometry data) [12] , milo [8] and ANCOM-BC [11]. Here, we first generated a large pool of single-cell transcriptomic profiles with a simulator Splatter [26] (Fig. 2A; Additional file 1: Fig. S4). This simulated pool was then used as the seed data and cells are randomly selected according to the simulated cell-type proportions (as ground truth; see more details in Method). Clustering was further performed on the subsets of simulated cells with Seurat to mimic the potential confusion introduced in the clustering step (the empirical confusion matrix shown in Additional file 1: Fig. S7). The seven methods were then performed on the cell counts obtained from the clustering annotation. Uniquely, we estimated the similarity matrix between cells using a KNN graph by calculating the fraction of neighbors for all cells in each cell type that

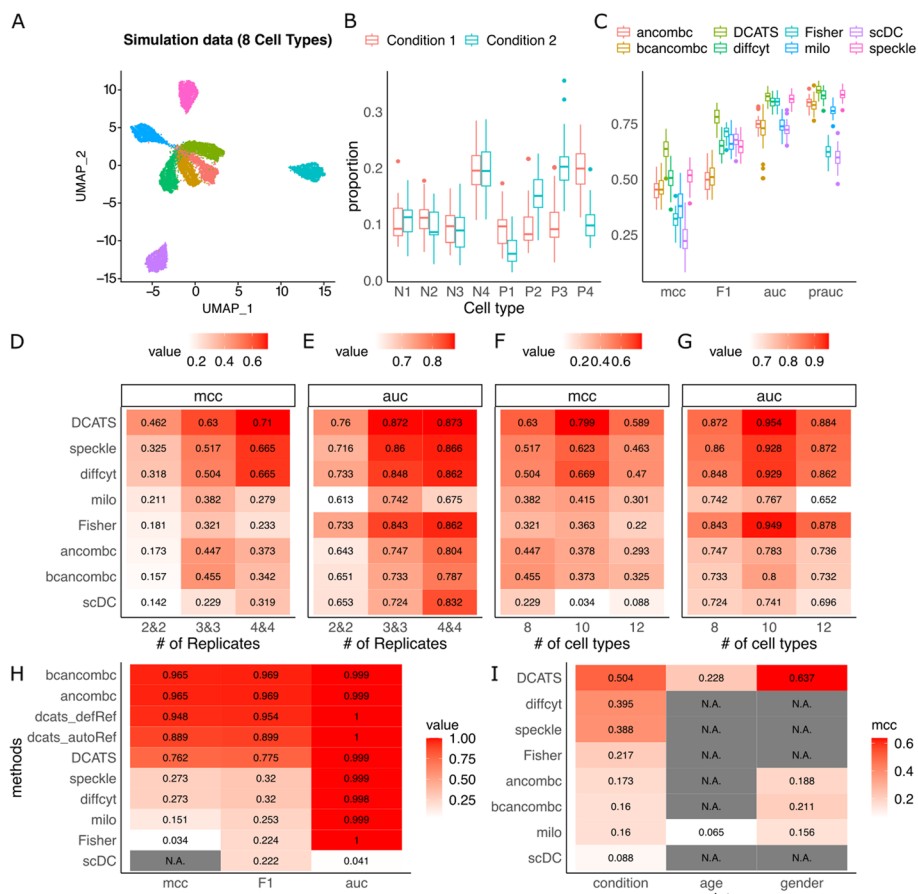

**Fig. 2** Evaluation of DCATS with multiple simulation datasets. **A** The UMAP plots of simulation data with 8 cell types. **B** The true proportions of different cell types in default setting (3 replicates in each condition across 30 runs; the 4 cell types with differential composition abundance have names started with "P"). **C** The boxplot of MCC, F1, AUC, and PRAUC of different methods in bootstrap sampling results with the default setting. **D**-**G** Comparing multiple methods with Splatter-simulated data by varying the number of replicates (**D**, **E**) and number of cell types (**F**, **G**). "bcANCOM-BC" indicates using bias corrected proportions estimated by DCATS as the input of ANCOM-BC [11]. The default number of cell types is 8, and the default number of replicates is 3. **H** The performance of each method in the special case. "DCATS" indicates the results of DCATS using total cell count as the normalization term. "dcats_autoRef" indicates using the reference group automatically detected by DCATS as the normalization term. "dcats_defRef" indicates using the know true reference group as the normalization term. **I** Detecting variable cell types with multiple covariates, for both categorical and continuous types. Data is simulated with Splatter. N.B., only DCATS and Milo support multiple covariates

belong to different cell types. It approximated the empirical confusion matrix well (Pearson's $R = 0.9987$; Mean absolute error $= 2.977 \times 10^{-18}$; Additional file 1: Fig. S8).

We generated three types of simulation data. Among all the simulation scenarios, we simulated multiple samples coming from two conditions. In the first type of simulation, we focused on the change of proportion. The proportion of some cell types increased, while the proportion of other cell types decreased. The proportion of samples coming from two conditions all summed up to 1. In the second type of simulation, we designed the simulation based on different cell counts of each cell type. With the count of one cell type increased from condition 1 to condition 2, the proportion of this cell type increased, while the proportions of the rest cell types decreased. In

the third type of simulation, we included another two confounding covariates - age and gender, to test the ability of controlling covariates of different methods.

For the first type of simulation data, in a default setting with 3 replicates in each condition and 4 out of 8 cell types with differential composition abundance (Fig. 2B), we found that DCATS outperforms all other methods (area under ROC curve (AUC): 0.872 vs 0.86; matthews correlation coefficient (MCC) : 0.63 vs 0.517; F1: 0.779 vs 0.714, respectively for DCATS and the second-best method in each metric; Fig. 2C–G and Additional file 1: Table S1, 3, Additional file 1: Fig. S5–6). N.B., DCATS remains the best-performed method in all three metrics while the second-best method varies among the alternative methods. By ablating the correction of misclustering in DCATS and introducing the over-dispersion term in the beta-binomial regression, we observed that both bias correction and global estimation of over-dispersion contribute to the improvement of overall performance (Additional file 1: Table S1-2, Additional file 1: Fig. S6). When varying the number of replicates to 2 or 4, or the number of cell types to 10 or 12, we found that DCATS consistently outperforms all other methods, mostly by a large margin (Fig. 2D–G, Additional file 1: Table S1-4, Additional file 1: Fig. S6). Unsurprisingly, the increase of replicates improves the performance for almost all methods, especially for DCATS, suggesting its capability of estimating biological variability from replicates. It is worth mentioning that despite Milo's unsatisfactory performance in cell type levels, as a tool designed for detecting perturbation of cell states in partially overlapping neighborhoods, it shows its strength in differential abundance analysis at neighborhood level compared to DCATS and speckle. (Additional file 1: Fig. S9–10)

By comparing the results of ANCOM-BC using the observed proportion (ancombc) and bias-corrected proportion estimated by DCATS (bcancombc), we found that the bias correction step always increased the sensitivity of ANCOM-BC (Additional file 1: Table S1,3). However, the specificity sometimes decreased, and this led to the overall similar performance of ancombc and bcancombc. While in DCATS, using bias-corrected proportions led to a higher increase in sensitivity and almost no decrease in specificity (Additional file 1: Table S1,3).

Next, we conduct the second type of simulation to assess the composition effects in the task. The first cell type has a 20 times count increase in condition 2, while counts of the rest cell types remain the same. Proportions of each cell type are calculated based on the defined cell counts, and the simulation is done based on these cell types' proportions. With the same total count in 2 conditions, the observed cell count of the first cell type shows a huge increase in condition 2, while cell counts of the rest cell types decrease (Additional file 1: Fig. S11). We test the performance of different methods using this dataset. With the estimated dispersion term, this systematic decrease of proportion can be partially modeled by DCATS (DCATS (estPhi_emK): 0.762 vs wtoPhi_emK: 0.190 in MCC, Fig. 2H, Additional file 1: Table S6). Thus, DCATS achieves higher MCC and F1 compared to other methods.

To directly address the issue of cell type proportion, we investigated the performance of selecting certain cell types as the reference group for normalization, rather than utilizing all remaining cell types (as outlined in detail in the "Method" section). Specifically, the reference cell types should exhibit no significant abundance differences with high confidence. In this simulation, employing three known unchanged cell types as the

reference group enabled DCATS to differentiate between actual abundance changes and spurious effects caused by other cell types (0.762 to 0.948 in MCC, 0.775 to 0.954 in F1). The performance of DCATS improved mainly through the increase in specificity (0.917 to 0.986, Additional file 1: Table S6). Such improvement is also seen when using 3 reference cell types automatically recommended by DCATS (0.762 to 0.889 in MCC, 0.775 to 0.899 in F1).

Moreover, a third simulation was performed to evaluate DCATS's ability to account for additional covariates or to jointly test multiple covariates in association with composition abundance for each cell type. Specifically, we simulated ten replicates for each condition with different ages and genders. There were a total of eight cell types, of which four displayed differential abundance between conditions. In addition, we selected four cell types to exhibit confounding effects from both age and gender. Two of these cell types exhibit differential abundance (Additional file 1: Table S5). Even though most of these seven methods are based on a linear model framework, DCATS, milo, and ANCOMBC are the only three methods designed with the utility to support additional covariates testing, while ANCOMBC only supports testing for discrete covariates. scDC, as a GLM based method, is in principle able to support covariates, but the interface is not implemented. propeller in the speckle package [10] only takes the condition information as input but not additional covariates. diffcyt, which uses edgeR [13] for differential abundance analysis on flow cytometry data also met challenges in incorporating different covariates due to its highly specific analysis pipeline designed for flow cytometry data. Therefore, we focus on Milo and ANCOMBC for incorporating covariates, while interestingly they fail to effectively control the influence of confounding covariates when performing the analysis (Fig. 2I, Additional file 1: Fig. S12–13). By contrast, DCATS achieves improved MCC partly thanks to its capability of jointly modeling additional covariates. Indeed, DCATS shows good performance in detecting cell types with additional covariates for both continuous (age) and discrete (gender) variables (Fig. 2I, Additional file 1: Fig. S12–13).

### Evaluating DCATS on experimental data sets

To further illustrate the performance of DCATS, we applied DCATS and five other methods (Fisher's exact test, scDC [14], speckle [10], milo [8], and ANCOM-BC [11]) to three experimental datasets. As we do not have the KNN matrices required for annotating cell types in these datasets, similarity matrices are calculated by performing classification with SVM over the original cell type annotation.

We first assessed the sensitivity of DCATS using a dataset (Angelidis2019) containing single-cell transcriptomic and mass spectrometry-driven proteomic data of whole lungs from 3-month-old mice ($n = 8$) and 24-month-old mice ($n = 7$) [27]. In this dataset, the authors discovered one cluster of cells with a high expression level of S and G2M cell-cycle marker genes. This cluster contains mainly proliferating cells, thus, should show higher abundance in young mice. As these proliferating cells belong to T cells, type-2 pneumocytes, and alveolar macrophages, these cell types should have higher proportions in young mice. Using single-cell transcriptome information, DCATS enabled the detection of type-2 pneumocytes, and alveolar macrophages, and a subtype of T cells, CD4+ T cells, as differentially abundant cell types (Additional file 1: Table S8). In addition, the

authors found a relatively increased number of ciliated cells in old mice compared to club cells. By deconvoluting bulk RNA sequencing data, the authors found the upregulation of ciliated cell marker genes signature. The increased ratio of ciliated to club cells in aged mice is further validated by immunostainings of Foxj1 (ciliated cell marker) and CC10 (club cell marker) [27]. Using club cells as the reference group, DCATS enabled the detection of this ratio change ($p = 0.0053$, Additional file 1: Table S8).

We then assessed the performance of these six methods on controlling false positives by applying them to a negative control data set on PBMCs, where no cell type was reported to have significant proportion change between 8 lupus patients treated with interferon(IFN)-$\beta$ and 8 control samples [4]. Here, we found that DCATS, speckle and ANCOM-BC both have good control of false positives, with no cell types having $p < 0.1$ (Fig. 3A). Similarly, Milo also has a good control on false positives when manually setting its threshold to 0.2 based on prior knowledge (this threshold is used throughout all data sets below; Additional file 1: Table S7). On the other hand, Fisher's exact test severely suffers from type I errors (4 out of 8 cell types passing a significance level at $p < 0.01$); scDC also returns three cell types with $p < 0.1$, including B cells with $p < 0.01$.

The third dataset consists of 53,193 epithelial cells from mice's small intestine and organoids [28]. It includes 4 control samples, 2 samples from two days after *Salmonella* infection, 2 samples from three days after *H. polygyrus* infection, and 2 samples from ten days after *H. polygyrus* infection. The original study defined eight cell types and compared between controls and each stimulation group to identify cell types with differential composition abundance through a Poisson regression and Wald test [28]. Presumably, since the sample variability was not well considered in the Poisson regression, the authors used FDR $< 1 \times 10^{-5}$ as the cutoff of differential abundant cell types between control and simulations.

We do not have the ground truth about which cell types are differentially abundant, but for the purpose of evaluating the accuracy of DCATS and other methods, we used the findings in the original study of this dataset as the "silver standard." When re-analyzing this data set, we first noticed that Fisher's exact test captured all the reported differential abundant cell types, but also reports lots of differential abundant cell types that are regarded as non-differential abundant cell types by other methods in each of the comparison pairs (with $p < 0.01$). Similarly, scDC and milo also returns "false positives" in each comparing condition pair but occasionally missed reported positives. Surprisingly, speckle shows limited power, missing almost all hits. In contrast, DCATS displayed a good balance between sensitivity and specificity; it does not return any "false positive" (using 0.05 as the significance level) and captures most of the reported differential abundant cell types when comparing the control group and *H. polygyrus* infection groups. When comparing the control groups and *Salmonella* infection group, we observed a significant increase in the proportion of enterocytes, which causes proportions decrease in the rest of the cell types. Using the reference group detected by DCATS as the normalization term was able to overcome the influence of enterocytes. The results show that enterocyte is the only cell type with differential abundance (Fig. 3B–D, Additional file 1: Table S9-10, Additional file 1: Fig. S14).

Furthermore, if we treat the results given in the paper as results of another method, we can use the major decisions as the highly possible conclusions. In this case, the

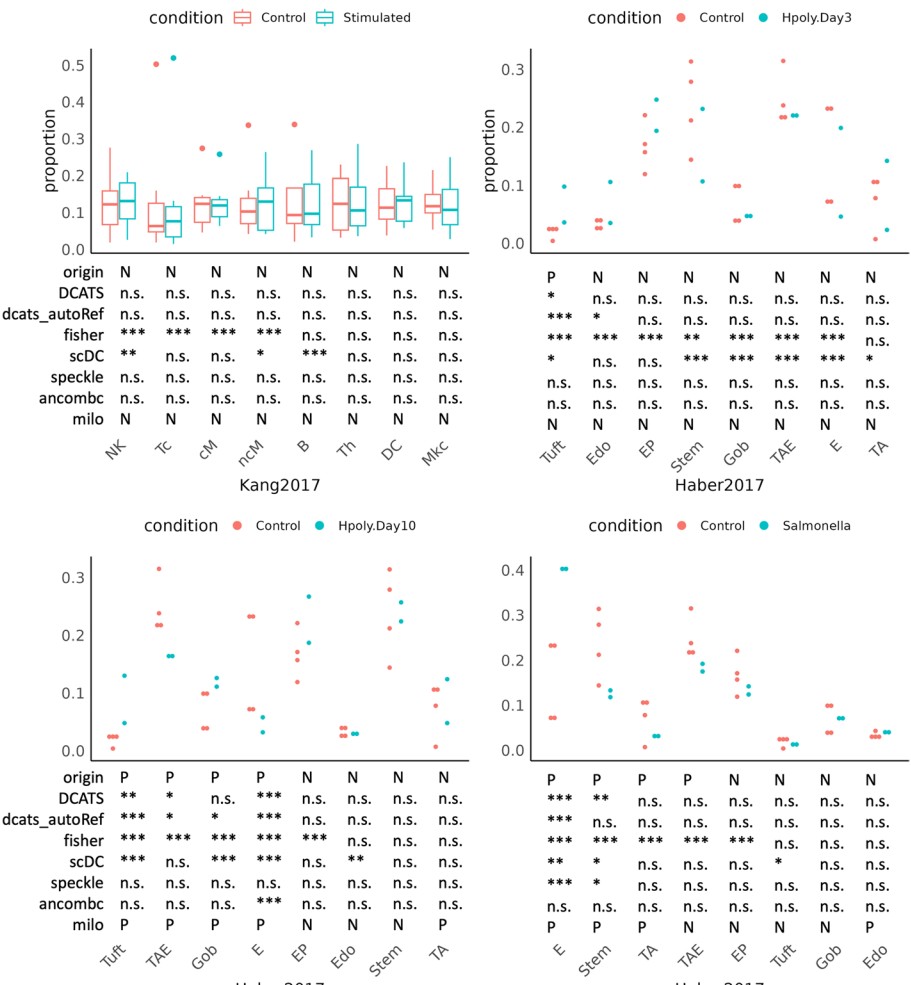

**Fig. 3** DCATS gives accurate conclusions for two experimental datasets. "dcats_autoRef" indicates using the reference group automatically detected by DCATS as the normalization term. **A** shown is the proportion of each cell types in the Kang dataset [4]. cM, CD14+CD16- monocytes; ncM, CD14+CD16+ monocytes; DC, dendritic cells; Mkc, megakaryocytes; Th, CD4+ T cells; B, B cells; Tc, CD8+ T cells; NK, natural killer cells. **B**–**D** show the proportion of each cell types in the Haber dataset [28]. E, Enterocyte; TA, transit amplifying; TAE, TA.Early; EP, Enterocyte.Progenitor; Gob, Goblet. "P" in the first line represents existing proportions' difference according to the original papers, and "N" represents no significant proportions' difference. "P" in the last line represents existing proportions' difference based on results of milo, and "N" represents no significant proportions' difference. In the rest four lines, "*" represents *p*-values from 0.05 to 0.1, "**" represents *p*-values from 0.01 to 0.05, "***" represents *p*-values less than 0.01. "n.s." means not significant

performance of each method is evaluated by their similarity with other methods. We can find that DCATS provides a consistent conclusion with the majority decisions of each cell type in different comparisons, except for the significant proportion change of Goblet between control and Hpoly day 10 conditions.

## Application of DCATS in complex experiment designs

To demonstrate the utility of DCATS for complex designs, we selected a cohort study including 196 hospitalized COVID-19 patients with moderate or severe disease, corresponding control group, and patients in the recovering stage. Within this cohort study, we selected scRNA-seq data from fresh and frozen PBMC samples with known

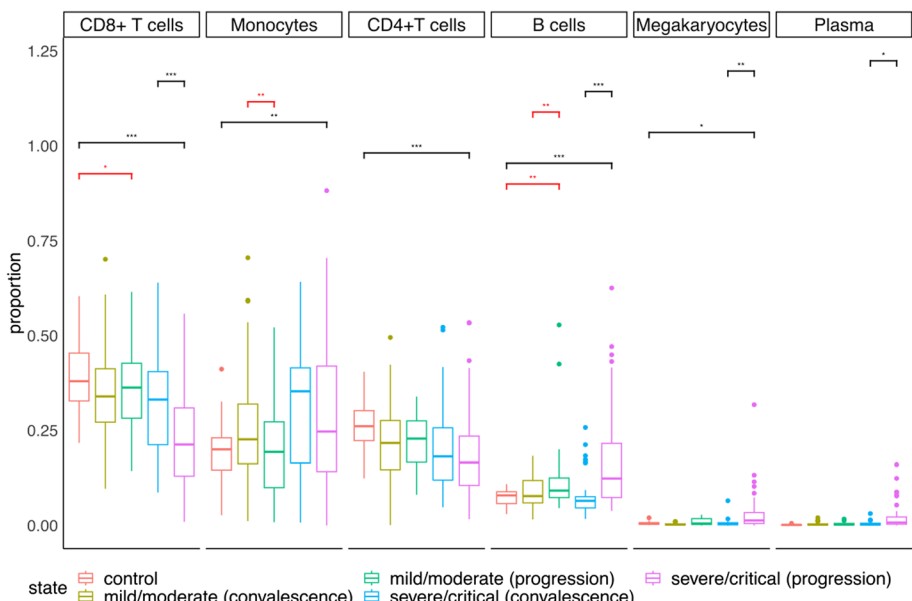

**Fig. 4** DCATS is able to find new cell types with differential abundance in Ren dataset [29]. Cell types' proportion and conclusions on the Ren dataset given by DCATS. The significant bars in black color indicate that results are the same as the original paper, while the significant bars in red colors indicate newly discovered cell types with proportion differences

metadata. The samples were stratified into five groups: 20 samples from the control group, 48 samples from the mild or moderate convalescence group, 19 samples from the mild or moderate progression group, 36 samples from the severe or critical convalescence group, and 48 samples from the severe or critical progression group [29]. In the original study, differential abundance analysis was performed between multiple severity groups.

Different from the original study, we leveraged DCATS's capability to account for confounding factors, including age and gender through the GLM framework. We noticed both DCATS and the original study identified a few cell types with substantial proportion changes (denoted in by black asterisks in Fig. 4), such as, CD8+ T cells between severe progression and convalescence or healthy controls. On the other hand, DCATS also uniquely detects several cell types that show a significant proportion difference (highlighted in red).

Specifically, DCATS indicates a slightly different proportions of CD8+ T cells between the control group and the mild/moderate progression group ($p = 0.067$). This is consistent with what another study found when analyzing high-dimensional cytometry data from 125 COVID-19 patients, corresponding recovered group, and healthy control group [30]. Even though the significance is moderate, the trend is strong. We also observed a significant difference in the proportion of monocytes between the mild/moderate progression group and corresponding recovery group ($p = 0.013$). This finding is consistent with a recent report from Qin and colleagues by using fluorescence-activated flow cytometry analysis to track the dynamic changes of monocytes in patients during the recovery stage from mild symptoms [31]. DCATS also shows a significant difference in proportion between the control group and the mild/moderate progression group

($p = 0.024$) as well as the mild/moderate progression group and corresponding recovery group ($p = 0.021$) regarding B cells. The same influence on B cells is also described in the original paper with a stronger signal given by DCATS between the control group and the severe/critical progression group ($p = 3.10 \times 10^{-4}$) as well as the severe/critical progression group and corresponding recovery group ($p = 2.48 \times 10^{-6}$). These new findings show that DCATS is a powerful tool in identifying cell types with significant proportion changes while keeping high specificity.

## Discussion

Comparing to simple statistical tests like Fisher's exact test, DCATS preserves high sensitivity while having a good performance in specificity. Through multiple simulation and experimental datasets, we demonstrated that DCATS has an excellent overall performance and outperforms existing methods. Strikingly, DCATS has strong control of false positives but maintains a high level of power.

When benchmarking with other methods, Milo shows the unsatisfactory performance when we conducted the differential analysis at cell type level. This is partly because that Milo is designed at the neighborhood level, and there is no build-in metric to determine the differential abundance at cell type level, hence the proportion of neighborhoods with significant difference were used as a surrogate to indicate cell-type level significance. Despite the unsatisfactory performance for Milo, it shows its strength in detecting differential abundance neighborhoods when compared to other methods, and is a good choice when we have data with continuous cell state changes.

As DCATS corrects the misclassification bias based on the similarity matrix, the estimation of this matrix is an important step and can influence the performance of DCATS. Although, we found that KNN-based or empirical classification-based similarity matrices function well in general, the estimation of this matrix may be further improved in the future, in the light of more benchmarking datasets with accurate cell type annotations.

Lastly, the differential abundance of cell type naturally regards relative abundance. Hence, choosing the reference (i.e., non-differential cell types) can potentially affect the analysis results. Typically, we assume that the cell type abundance change is relatively small, hence using all remaining cell types is our default setting and demonstrate robust performance in broad scenarios. On the other hand, if the compositional change is substantial, all non-differential cell types will be affected and selecting certain cell types as a confident reference can reduce such impact, e.g., by DCATS's automatic recommendation. Nonetheless, such a large compositional difference is not a typical scenario to apply DCATS and we suggest users perform the reference cell selection with caution.

## Conclusions

In this works, we introduced DCATS, in a form of beta-binomial regression, for detecting cell types with differential composition abundance. With the bias correction procedure based on a similarity matrix, DCATS provides a robust framework for differential composition analysis between conditions. As the input of DCATS is the count matrix which includes the number of cells in each cell type for each sample, it can deal with predefined cell type annotation, and can be easily adapted to any other single cell (multi-) omics data. This unique feature also allows DCATS to be implemented on data with

self-adjusted annotation based on integration of multi-omics data or biological knowledge. We demonstrated the usability of DCATS in both simulated data set and real world data set.

## Method

### Composition correction with clustering similarity matrix

From a scRNA-seq experiment, we can obtain the cell counts $x = \{x_1, ..., x_K\}$ of $K$ cell types by performing cell clustering, e.g., Louvain method [6] and K-means. If the clustering is unbiased and accurate, it is natural to take the cell type composition abundance $z$ proportional to the returned cell count vector $n$. However, due to the complex underlying structure, the clustering step is often inaccurate and may be systematically biased, hence it is crucial to correct the bias on composition abundance introduced by clustering method. In next subsection, we demonstrate that the clustering similarity matrix $M$ (or similarity matrix) can be well approximated, e.g., by KNN graph. Specifically, the element $m_{i,j}$ denotes the probability of a cell $c$ in type $i$ assigned to type $j$ by the clustering method, namely, $P(A_c = j | I_c = i) = m_{i,j}$, where $A_c, I_c$ respectively denote the clustering assignment and genuine identity of cell $c$. This also means that $M$ could be asymmetric and requires $\sum_{k=1}^{K} m_{t,k} = 1$ with $m_{t,k} \geq 0$ for any cell type $t$.

Given the above definition and a pre-defined clustering similarity matrix $M$, we could have the likelihood of unknown cell type composition vector $z$ on observing the cell counts $n$, as follows,

$$
\begin{aligned}
\mathcal{L}(z) = P(n|z, M) &= \prod_{j=1}^{K} \prod_{c=1}^{n_j} \left[ \sum_{i=1}^{K} P(I_c = i|z) P(A_c = j | I_c = i) \right] \\
&= \prod_{j=1}^{K} \prod_{c=1}^{n_j} \left[ \sum_{i=1}^{K} z_i m_{i,j} \right] = \prod_{j=1}^{K} \left[ \sum_{i=1}^{K} z_i m_{i,j} \right]^{n_j} = \prod_{j=1}^{K} v_j^{n_j},
\end{aligned}
\tag{4}
$$

which is equivalent to a multinomial distribution parametrised by an adjusted composition abundance vector $v = z \times M$.

Algebraically, we could take $z = M^{-1} \times v$ with inverting the similarity matrix $M$. However, this simplistic treatment does not guarantee that the solution satisfies with $z_t \geq 0$ and $z_t \leq 1$ for any cell type $t$. In practice, constraints have been introduced to address this optimisation problem, e.g., [15, 16], though it occasionally returns unstable solutions.

Here, we introduce an Expectation-Maximization (EM) algorithm to achieve a maximum likelihood estimate of the cell type composition $z$ by introducing $\mu_{j,i}$ as the expected probability of clustered cell type $j$ coming from the real cell type $i$. Though this auxiliary variable $\mu_{j,i}$ functions like the inverse similarity matrix $M^{-1}$, its interpretation and calculation are different. Here, we could interpret $\mu_{j,i}$ as the posterior probability of clustered cell type $j$ coming from the real cell type $i$, hence can be calculated in a forward way (i.e., the E step) if given an estimated the true composition vector $z$ in a previous step, as follows,

$$\mu_{j,i} = P(I_c = i|A_c = j) = \frac{P(A_c = j|I_c = i)P(I_c = i|\mathbf{z})}{\sum_{t=1}^{K} P(A_c = j|I_c = t)P(I_c = t|\mathbf{z})} = \frac{m_{i,j}z_i}{\sum_{t=1}^{K} m_{t,j}z_t} \tag{5}$$

Within the EM algorithm, once we have the auxiliary variable $\boldsymbol{\mu}$, we can maximize the log likelihood in Eq. (4) by taking its derivative as zeros for each cell type, together with a Lagrange multiplier on the constraint $\sum_{k=1}^{K} z_k = 1$. By solving these equations (i.e., M-step), we can obtain an updated $\mathbf{z}$ with $\boldsymbol{\mu}$ calculated from a previous step.

$$z_i = \frac{\sum_{j=1}^{K} \mu_{j,i}n_j}{\sum_{t=1}^{K} \sum_{j=1}^{K} \mu_{j,t}n_j}. \tag{6}$$

Therefore, by alternating the E step in Eq. (5) and M step in Eq. (6), we can achieve a maximum likelihood estimate of corrected cell type composition vector $\mathbf{z}$ once the log likelihood stops increasing.

### Estimation of clustering similarity matrix

As mentioned above, the clustering of single-cell is often inaccurate with systematic bias. So we introduce a similarity matrix to describe this kind of misclassification bias. We assume that this kind of bias comes from the similarity between different cell types and variety within each cell type. Thus, the conditions of samples where our single-cell data comes from do not influence the direction of this bias. In other words, misclassification error introduced by the clustering process is consistent across all samples coming from different conditions.

Currently, DCATS supports three strategies for estimating the confusion matrix: Uniform, KNN and classification. Firstly, the uniform confusion matrix is defined as

$$\mathbf{M}^{(u)} = (1 - \epsilon(K + 1)/K)\mathbf{I} + \epsilon/K \tag{7}$$

where $K$ is the number of cell types; $\mathbf{I}$ is an identify matrix and $\epsilon$ is the level of confusion, which are set to 0.05 by default. It describes an unbiased misclassification across all the cell types. It means that one cell from a cell type have equal chance to be assigned to rest other cell types.

Secondly, the KNN-based similarity matrix is estimated from the knn graph based on the transcriptome, e.g., provided by Seurat [23]. It calculates the proportion of neighborhoods that are regarded as other cell types. In this case, DCATS corrects cell proportions mainly based on the information of similarity between different cell types and variety within each cell types. By defining $n_{x,j}$ as the number of neighbors for cell $x$ that are classified as cell type $j$, we can calculate KNN-based similarity matrix, e.g., its entry $m_{i,j}$ for similarity (misclassification) from cluster $i$ to $j$ as follows:

$$m_{i,j} = \frac{\sum_{x=1}^{x_i} n_{x,j}}{\sum_{j=1}^{K} \sum_{x=1}^{x_i} n_{x,j}}$$

We realize in some situations, the cell clustering process might not be performed using the Seurat pipeline. It is also a common situation where people perform the clustering process based on part of the markers which they are interested in. With the development of technology, more information can be used for single cell clustering including spatial

information as well as biophysical feature. In above cases, knn matrix given by the Seurat pipeline cannot fully capture the bias introduced through the clustering process.

Based on the above reason, we introduce the third way to estimate a similarity matrix. The input for estimating the similarity matrix will be a data frame containing a set of variables that the user believe will influence the result of the clustering process as well as the cell type labels for each cell. We then use 5-fold cross validation and use support vector machine as the classifier to estimate the similarity matrix.

## Detecting differential composition abundance

### Generalized linear model

We assume the corrected cell counts $z_{s,j}$ for cell type $j$ given sample $s$ follow beta-binomial distribution. We then describe $z_{s,j}$ with a beta-binomial generalized linear model(GLM) with a logit link function for each covariate $i$, as follows,

$$
\begin{aligned}
z_{s,j} &\sim \texttt{Binom}(n_s, p_{s,j}) \\
p_{s,j} &\sim \texttt{Beta}(\alpha_{s,j}, \beta_{s,j})
\end{aligned}
\tag{8}
$$

where we reparameterized with the mean $\bar{p}_{s,j}$ and over dispersion $\phi_j$ as $\alpha_{s,j} = \bar{p}_{s,j}(1/\phi_j - 1)$ and $\beta_{s,j} = (1 - \bar{p}_{s,j})(1/\phi_j - 1)$. As described below, the mean parameter $\bar{p}_{s,j}$ is regressed to a set of covariates under different hypothesis. The weights for covariates and $\phi_j$ will be optimized to achieve maximum likelihood, through the *aod* package. The $p$ value can be calculated with a likelihood ratio test by comparing the log likelihoods in both alternative and null hypotheses.

*Full mode and Null mode* In DCATS, we allow using both "full mode" and "null mode" to finish the differential abundance analysis. When using null mode, we compare these two models:

$$
\begin{aligned}
\mathbf{H_0} &: \quad \texttt{logit}(\bar{p}_{s,j}) = w_0 \\
\mathbf{H_1} &: \quad \texttt{logit}(\bar{p}_{s,j}) = w_0 + w_i \times c_{s,i}
\end{aligned}
\tag{9}
$$

In this case, we only consider the influence of the factor we are evaluating. When using full model, we compare these two models as follows,

$$
\begin{aligned}
\mathbf{H_0} &: \quad \texttt{logit}(\bar{p}_{s,j}) = w_0 + \boldsymbol{w}^\top \boldsymbol{c}_s; \ w_i = 0 \\
\mathbf{H_1} &: \quad \texttt{logit}(\bar{p}_{s,j}) = w_0 + \boldsymbol{w}^\top \boldsymbol{c}_s
\end{aligned}
\tag{10}
$$

where $\boldsymbol{c}_s$ and $\boldsymbol{w}$ are respecitvely for the weight and covariate vectors for sample $s$. Under the null hypothesis, the $w_i$ will assumed to be 0. In this case, we test the differential abundance of different condition controlling the influence of other confounding covariates.

### Fixed overdispersion term

We found out that when the numbers of biological replicates are low, the over-dispersion term $\phi$ can be over-estimated. Thus, DCATS allows estimating a global over-dispersion term across all cell types, where we assume that the over-dispersion $\phi_j$ remains the same across for any cell type $j$. To estimate a global dispersion term, we fit all cell types in one joint beta-binomial GLM by taking the design matrix as the cell

type indicators. Then, the dispersion term estimated here is treated as a global dispersion and will be taken as a prefixed value when performing the main round GLM for each cell type.

Except for the mode we used in main text (Using KNN matrix as the similarity matrix and fixing the overdispersion term), three other modes of DCATS are tested in the simulation data (Additional file 1: Fig. S2, Additional file 1: Fig. S4 , Additional file 1: Fig. S7–11).

- "wtoPhi_wtoEM" indicates using basic beta-binomial distribution without bias correction or fixing over-dispersion term.
- "wtoPhi_emK" indicates without fixing the over dispersion term Phi but clustering bias corrected by KNN matrix.
- "wtoPhi_emU" indicates without fixing the over dispersion term Phi but clustering bias corrected by uniform matrix.
- "estPhi_wtoEM" indicates using fixing the over dispersion term Phi but no clustering bias correction.
- "estPhi_emK" indicates using fixing the over dispersion term Phi and clustering bias corrected by KNN matrix.
- "estPhi_emU" indicates using fixing the over dispersion term Phi and clustering bias corrected by uniform matrix.

**Use reference cell types for normalization**

When one cell type has extreme proportion change, the proportions of the rest cell types will be largely influenced, which might lead to false positives of differential abundance testing. Using known unchanged cell types as reference groups for normalization will offset the influence of massive cell types change. When using the total cell number as the normalization term, we have

$$\texttt{logit}(E(\frac{z_{s,j}}{n_s})) = w_0 + \boldsymbol{w}^\top \boldsymbol{c}_s \tag{11}$$

where values in $\boldsymbol{w}^\top$ could be zero based on different models. When using the reference group as the normalization term, we have

$$\texttt{logit}(E(\frac{z_{s,j}}{\sum_{r \in R} z_{s,r}})) = w_0 + \boldsymbol{w}^\top \boldsymbol{c}_s \tag{12}$$

where $R$ denotes the reference group.

We recommend using more than one cell type and more than 25% of total cells as the reference group to guarantee the stability of the normalization term.

DCATS also support the recommendation of the cell types used for reference. Namely, DCATS first calculates the $p$-values for each cell type. Then, it orders the cell types from high to low based on the $p$-values and provides the possible reference cell types based on this order. Additionally, DCATS calculates the proportion of cell counts for different numbers of reference cell types and suggests a recommended number.

### Theoretical simulation

The input of DCATS is combined with two parts. The cell count matrix $C_i$ denotes numbers of cells we observe for each cell type in each sample coming from same condition $i$. For each analysis, DCATS can only compare cell count matrices coming from two condition. Another part is the similarity matrix $M$ we describe above.

We first simulated two count matrices $C_1, C_2$ directly to demonstrate that introducing the similarity matrix to correct the unknown cell type composition vector $\boldsymbol{\mu}$ is important for the differential composition analysis. The numbers of replicates for condition 1 and condition 2 were 2 and 3 respectively. The total cell numbers were 2000 for each sample in both condition. We defined the genuine proportions of the three cells types were [1/3, 1/3, 1/3] and [1/3, 1/2, 1/6] in conditions 1 and 2, respectively. Using genuine proportions times 70 as concentration parameters, we generated the unknown cell type composition vector of condition 1 and condition 2 from Dirichlet distribution. For each sample, the total cell number 2000 and $\boldsymbol{\mu}_i$ were used to generate a cell count vector from multinomial distribution. Based on the similarity matrix we designed (Fig. 1B), the misclassification errors were introduced by reassigning cells into different cell types.

Next, we tried to demonstrate the effectiveness of the bias correction step. We first used DCATS and Fisher's exact test to perform the differential composition analysis. We then used the similarity matrix we designed to correct the miscalssification error we introduced. In this case, we used the true similarity matrix to do this bias correction process. For each simulation, we repeated the above process 50 times and calculate MCC, sensitivity and specificity. Overall, we got 50 Matthews correlation coefficient (MCC), sensitivity and specificity for 50 simulation.

### Simulation with single-cell RNA gene count matrices

When analyzing single cell RNA sequencing (scRNA-seq) data, we always start from gene expression matrix. Here, we designed a simulation process to mimic the process from getting gene expression matrix to analyzing differential compositions. We first used splatter [26] to generate a large cell pool including different cell types. We then generated a proportion vector $\boldsymbol{p}_s^{(i)}$ for each sample $s$ from condition $i$ following a Dirichlet distribution. Defined genuine proportions were used as concentration parameters. Given a total cell number, a cell count vector $\boldsymbol{\mu}_s^{(i)}$ was generated from multinomial distribution.

In a default setting, each condition contained 3 replicates, and 4 out of 8 cells types with differential composition abundance. The genuine proportion was [0.1, 0.1, 0.1, 0.1, 0.1, 0.1, 0.2, 0.2] in condition 1 and [0.05, 0.15, 0.1, 0.1, 0.1, 0.2, 0.2, 0.1] in condition 2. For each sample, the total cell number was 3000. We also changed the number of replicates to 2 and 4, and the number of cell types to 10 and 12. When the number of cell types equalled to 10, the genuine proportions were [0.05, 0.05, 0.1, 0.1, 0.1, 0.1, 0.1, 0.05, 0.2, 0.15] and [0.1, 0.1, 0.1, 0.1, 0.1, 0.1, 0.1, 0.1, 0.1, 0.1] in condition 1 and condition 2, respectively. When the number of cell types was 12, the bass proportions were [0.1, 0.1, 0.1, 0.1, 0.05, 0.05, 0.05, 0.15, 0.05, 0.05, 0.1, 0.1] and [0.1, 0.1, 0.1, 0.1, 0.1, 0.1, 0.1, 0.1, 0.05, 0.05, 0.05, 0.05].

Based on the cell count vector $\boldsymbol{\mu}_s^{(i)}$, we randomly selected cells as well as their gene expression profiles. We can therefore got a gene expression matrix for each sample. Then, we used Seurat [23] pipeline to do the pre-processing as well as the clustering

process. The cluster annotation of each cell was treated as input for the downstream analysis. We compared DCATS with other five tests designed for differential composition analysis which are Fisher's exact test, scDC [14], diffcyt [12], milo [8], and speckle [10] (Additional file 1: Fig. S1).

As milo aims to detect cell stage perturbations based on k-nearest neighbor graphs but not cell type levels, we also did the differential abundance testing on the neighborhoods level by assuming all the neighborhoods were influenced by condition if that cell types have differential proportions. In this case, we only considered neighborhoods having more than 80% of cells coming from one cell type and used neighborhoods detected by milo for testing using Fisher's exact test, speckle [10] and DCATS. For each scenario, we pooled the results coming from 30 times simulations together to calculate the F1 score, MCC, AUC and area under the precision-recall curve (PRAUC). The confidence interval of F1 score, MCC, AUC and PRAUC were calculated by bootstrapping the testing results 50 times.

### Adding other confounding covariates

To demonstrate that DCATS can control the influence of confounding covariates, we designed a simulation with eight cell types. First four of them had different proportions in different conditions, and rest four of them had same proportions in different conditions. We also added two confounding covariates, age and gender which influenced the proportions of four among eight cell types. Using 15 years old as the baseline, one year increase in age led to 0.05 decrease of concentration parameters in cell type 1 and cell type 5. It also led to 0.05 increase of concentration parameters in cell type 2 and cell type 6. Comparing to female, male had 2 increase of concentration parameters in cell type 2 and 6, 2 decrease in cell type 1 and cell type 5. (Additional file 1: Table S5) The gender for each sample was randomly selected from female or male. The age was randomly select from 15 to 45. The influence of condition, age and gender was additive and result in the final proportion of each cell types in each biological replicate. Overall, we had 10 biological replicates in each condition with different ages and genders. We simulated proportions based on the true designed proportions from a Dirichlet distribution and cell numbers based on the simulated proportions from the multinomial distribution. Based on the simulated cell counts, we simulated gene expression matrices and did the differential abundance analysis as described previously.

### Simulation starting from cell counts differences

To demonstrate that DCATS can handle spurious effects arising from compositionality characteristics — namely, when the proportion of one cell type increases, the proportions of the remaining cell types decrease — we conducted a simulation scenario where one cell type had differential abundance with increased counts, while the remaining cell types had the same counts. In this case, the remaining cell types had lower proportions than the original composition, but these proportion differences were merely side effects and should not be detected as differential abundance.

We simulated eight cell types with count units [1, 4, 4, 4, 4, 4, 4, 4] in condition 1 and [20, 4, 4, 4, 4, 4, 4, 4] in condition 2. The first cell type had a 20-fold count increase in condition 2, while the remaining cell types remained the same. We calculated the

proportions of each cell type and used these proportions for the simulation. The proportion of each cell type in condition 1 was [0.034, 0.138, 0.138, 0.138, 0.138, 0.138, 0.138, 0.138], while the proportion of each cell type in condition 2 was [0.417, 0.083, 0.083, 0.083, 0.083, 0.083, 0.083, 0.083]. We simulated three replicates for each condition. When the total count remained at 3000 in both condition 1 and condition 2, the observed cell count of the first cell type showed a huge increase in condition 2, while the cell counts of the remaining cell types decreased (Additional file 1: Fig. S11). The cell selection and testing procedures were the same as in the above section of simulating cell counts matrices.

### Methods for benchmark
#### *Fisher's exact test*
As Fisher's exact test does not support multiple replicates, we added the cell counts numbers of all replicates in each condition and tested based on the sum of cell counts.

#### *diffcyt*
diffcyt [12] is a package designed for differential discovery analyses in high-dimensional cytometry data. In order to adopt it to analyze scRNA-seq data. We first create data template using random generated flowset data, then replaced it with meta-data of data we simulate. We used default for the rest setting.

#### *ANCOMBC*
ANCOMBC [11] is designed for differential abundance analysis for microbiome data. We use the count matrix and design matrix to create TreeSummarizedExperiment objects [32]. These objects are used as input for ANCOMBC. Rest parameters are set as default. The count matrices we used include the count matrix calculated from Seurat clustering results (ancombc), and the bias-corrected count matrices calculated from DCATS (bcancombc).

#### *milo*
milo [8] aims to detect cell state changes in neighborhoods level, while both our simulated data and real-world data only have ground truth in cell type level. Thus, we linked neighborhoods with cell type based on the majority cell in that neighborhoods as what milo's authors did in the milo paper [8]. We first used FDR<0.1 as threshold to define whether one neighborhood has cell state changes, and calculated how many more neighborhoods one condition has dominant size comparing to the other condition as well as the proportion of this difference among all neighborhoods that belong to that cell type. This proportion is regarded as perturbation level. Inferring from the Kang dataset [4] which works as the negative control, we defined that one cell type shows significant proportion changes when the perturbation level is larger than 20%. Due to the less variety of simulated data, this threshold is setted to be 0 in simulated data in order to get better performance of milo.

### speckle and scDC

For speckle [10] and scDC [14], we used default setting and standard pipeline as described in the github page/tutorials of these two tools.

## Supplementary information

---

**Additional file 1.** Supplementary Table S1-S10 and Supplementary Figures S1-S14. Additional file 1 contains Supplementary Figures and Supplementary Tables.

**Additional file 2.** Review history.

---

**Acknowledgements**
We thank members in Ho lab, Huang lab from the University of Hong Kong, and Xuegong Zhang lab from Tsinghua University, especially Aaron Kwok, Weizhong Zheng, Ken Yu, and Junyi Chen, for fruitful discussions.

**Review history**
The review history is available as Additional file 2.

**Peer review information**
Andrew Cosgrove and Anahita Bishop were the primary editors of this article and managed its editorial process and peer review in collaboration with the rest of the editorial team.

**Authors' contributions**
YH and JWKH conceived the ideas and designed the study. XL developed the method, implemented the R package, and performed all the experiments. CC supported the initial development of the method, collected some data, and supported evaluation. KM performed some simulation experiments and analysis. XL, YH, and JWKH wrote the paper. All authors read and approved the final version of the manuscript.

**Funding**
This work was supported by AIR@InnoHK administered by Innovation and Technology Commission.

**Availability of data and materials**
All datasets used in the paper are previously published and can be found in NCBI GEO database: GSE96583 [4], GSE92332 [28], and GSE158055 [29].
 This algorithm is implemented as an R package named DCATS which is available at Github [33] under the MIT license. The codes for simulation and analysis are avalible in Github [34]. The source code used in this paper is deposited in Zenodo with DOI: 10.5281/zenodo.7969592 [35].

## Declarations

**Ethics approval and consent to participate**
Not applicable.

**Consent for publication**
All authors have approved the manuscript for submission.

**Competing interests**
The authors declare that they have no competing interests.

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

## 