## [**Additional file 2.** Review history. · Genome Biology]

Review History

First round of review

Reviewer 1

Were you able to assess all statistics in the manuscript, including the appropriateness of statistical tests used? Yes: See detailed comments in Major points.

Were you able to directly test the methods? Yes

Comments to author:

The authors present DCATS, a new method for differential abundance (DA) testing of cell type population data derived from single-cell RNA sequencing (scRNA-seq). The method consists of two steps: first, a corrected abundance table is computed trying to fix clustering errors by leveraging a dissimilarity matrix. Then, differential abundance testing with a Beta-Binomial regression model is performed on the corrected abundances. The method achieved good performances on simulated data and was also tested on real scRNA-seq datasets.

However, the authors' contribution is only marginal, as Beta-Binomial models for abundance data have been proposed and discussed before also in the context of single-cell data (Büttner et al., 2020; Martin et al., 2020).

Furthermore, the model disregards the compositionality of the data and it is not clearly demonstrated that the abundance correction step improves the cell type classification and therefore DA testing.

Also, we are not satisfied with the detail of description and scope of experimental setups, that a) do not allow to properly judge the method, and b) are insufficient to fully reproduce the experiments.

Major comments:

1) Although single-cell abundance data are clearly compositional and the authors acknowledge this at multiple occasions in the manuscript (sum-to-one constraint in M , Dirichlet-Multinomial sampling in the benchmark), they completely miss this property when testing for differential abundance. As shown by (Büttner et al., 2020) in their scCODA manuscript, a Beta-Binomial model suffers from the same false-positive inflation properties as the other tests used in the benchmarking setup (t-tests in speckle, Negative Binomial model in milo, ...). Therefore, it would be important to include also statistical tests that model compositionality into the benchmark. While we know of no compositional DA testing methods that incorporate cell type uncertainty, we assume that the adjusted cell count matrix from DCATS could be used as the input to a compositional method like ANCOM-BC (Lin and Peddada, 2020) to have a fair comparison solely between the DA testing steps. Such a comparison is necessary to ensure that the second part of the method does not suffer from flaws that are masked by the improvements gained by cell count adjustment.

2) The authors fail to show that the clustering bias corrections used in DCATS give an improvement in classification when compared to the cell type composition of a real biological sample. All that is achieved by adjusting the abundance table with the help of a clustering similarity matrix is to smooth the abundance data towards the results produced by another classification method (KNN or SVM). It is necessary to validate with real data whether this step actually reduces the errors in cell type clustering/classification produced by, e.g., Louvain clustering.

3) It is unclear whether the improved performance of DCATS stems from the correction of misclassifications in the abundance data or from a better DA testing model. Therefore, it is necessary to compare the methods before and after applying the bias correction step from DCATS, wherever applicable, to distinguish whether the bias correction or the Beta-Binomial model are the main sources of improvement.

4) There are some passages in the benchmark description that were not clear to us. First, the description of the theoretical simulation (Section 4.4) misses many important details (e.g., number of replicates, sample size, numerical values) or it is not clear from the description what is meant (What are 1S, 2S and how are 2 and 3 vectors simulated from them? Are those the sample sizes?).

The same is true for Section 4.5 (What is the range of total cell counts per sample? Is the base composition coming from splatter? How big are the differential effects?)

Further, the model comparison study does not disclose the number of replicates. Multiple datasets should be simulated for each sample size and dataset size to avoid biases. It is not clear from the methods section if the benchmark was replicated.

Finally, the description of the simulation with confounders does not disclose which cell types were affected by the confounders (presumably the same as the ones affected by the condition, but this is not clear) and whether the simulations were performed with one confounder (age/gender) each or also with both confounders at the same time. Also, information on effect sizes and value distributions of the confounders is missing.

Also, it is necessary to see a simulation where all DA cell types are increasing, to showcase that DCATS can deal with the spurious effects arising due to the compositionality of the data.

5) In the real data applications, it is dangerous to take the results reported by the original publications as a "ground truth". These results were obtained with inadequate statistical models, which the authors correctly point out (e.g., for the Haber et al. dataset). Therefore, simply obtaining the same results as the initial publication is not a good measurement of method performance, and only points out that the method may suffer from the same flaws. Because of this, we think that a judgment of methods based on their similarity with each other is more effective than comparing to one (likely inaccurate) result.

6) When showing performance indicators obtained on multiple replicates like F1 score or MCC, some sort of confidence interval or standard deviation should be reported. These values are missing in the entire manuscript.

Due to the small sample sizes, confidence intervals will be relatively large. While this is always true for testing on datasets with low replicates, it still has to be discussed.

Minor comments:

1) The notation between Sections 4.1 and 4.3 is inconsistent. x is used for both cell abundances and covariates, while μ is used for both the relative abundances and the Beta-Binomial mean. Please make sure that the notation is easily understandable and all values are defined.

2) Some heatmaps (Fig. 2D and S4) are color-coded by the method rank instead of the numerical value of the metric, which can be very misleading. If possible, the values of the metric should be used as a color scale.

3) On page 9, there is a typo: Instead of "Language multiplier", it should read "Lagrange multiplier".

References

1. Büttner, M., Ostner, J., Müller, C. L., Theis, F. J., and Schubert, B. (2020). scCODA: A Bayesian model for compositional single-cell data analysis. bioRxiv, 2020.12.14.422688. doi:10.1101/2020.12.14.422688.
2. Lin, H., and Peddada, S. D. (2020). Analysis of compositions of microbiomes with bias correction. Nat. Commun. 11, 3514.
3. Martin, B. D., Witten, D., and Willis, A. D. (2020). MODELING MICROBIAL ABUNDANCES AND DYSBIOSIS WITH BETA-BINOMIAL REGRESSION. Ann. Appl. Stat. 14, 94-115.

Reviewer 2

Were you able to assess all statistics in the manuscript, including the appropriateness of statistical tests used? Yes, see review

Were you able to directly test the methods? No

Comments to author:

Lin and colleagues present an interesting new tool called DCATS for differential abundance analysis that under the hood uses beta-binomial modelling as well as some bias correction. Overall, the paper reads nicely and is well-organised. I have a small number of comments that I think would be worth clarifying or addressing.

1. There is mention in multiple places near the Introduction about 'complex experimental design', 'between two conditions'. But, existing frameworks handle all this and have done so for years. So, I find this aspect of motivation somewhat strange (related comment below).
2. Why does Fisher's exact test get such a big mention? It's quite clear that this is not even a reasonable baseline, because it is not capable of handling replicates. It would've seemed more logical (especially for a statistics paper) to have text at the beginning to say that Fisher's exact is not appropriate; end of story. I would suggest to make a clear statement about the appropriateness of FET in this context.
3. Later in the paper, propeller and diffcyt are tested against DCATS. So, I find it surprising that these approaches are not mentioned in the introduction. Usually in an introduction, the entire space of relevant methods would be spelled out to the reader, not just a subset. I would suggest to expand the Introduction to have complete coverage of the methods in this space.
4. From a brief look, the R package looks well designed and documented, but I would suggest also that the authors submit their package to Bioconductor. This typically results in an

increase in the standard of packages and has some extra requirements in terms of interoperability with other packages, etc. It also has the benefit of additional exposure.

5. For me, in a methods paper, the result is the method. Thus, in the first section of the Results, I would suggest to give more mathematical details of the method, including the parameterisation of the BB model and a few more details of the error correction / similarity matrix calculation (in the second section of the Results). In addition, the papers suggests that DCATS has 2 novel features, but I'm not sure if either are really "novel" (I also think the authors don't need to "sell" the novelty; they propose a clever and reasonable approach and show that it works well; I would always suggest to focus on the results, not the salesmanship). For example, BB modelling has been used successfully in DNA methylation analysis for a long time:

<https://bmcbioinformatics.biomedcentral.com/articles/10.1186/1471-2105-15-215>

Perhaps also worth mentioning is their bias correction has similarity to that done in other 'differential abundance' contexts, such as microbiome analysis:

<https://www.nature.com/articles/s41467-020-17041-7>

(Also, there is a lot of literature on more-or-less the same topic in microbiome analysis that is not mentioned in the Introduction. Also, there is a lot of literature on 'compositional' analysis that is not well-covered in the Introduction.)

Also, given that "composition" is mentioned several times in the paper, it would make sense to compare their method against a compositional data analysis method, such as ANCOM-BC.

6. I think we need more details about how other (competing) methods were run. The statement "[Method X] fails to control influence of confounding covariates although we included related information when performing the analysis" is very surprising. Several of these methods are based on linear modelling frameworks such that they can adjust for covariates. And they magically all fail to do this? I find this suspicious. At the very minimum, the authors should make all the code available that they used to run these methods. But also some detective work would be nice to understand why all other methods fail. It simply cannot be that diffcyt, speckle, milo are completely unable to account for covariates (Figure 2E).

7. Scenarios: One thing that is noticeable from the simulations is that the ground truth changes introduced are offsetting. That is, some subpopulations increase in relative abundance, while others decrease. This is convenient, because it helps make the normalisation easier. But, what about a situation where one cell type expands in magnitude? (this is actually a common situation in immunological contexts). Here (i.e., one subpopulation expanding in magnitude), the normalisation is more challenging, because in a relative sense, all subpopulations now change (unless it's normalised out). Would DCATS handle such a situation?

8. In Figure 2C, the lines are impossible to see unless you go to 300% magnification on a large screen. I would also suggest to colour the x with the corresponding method. I also wonder whether ROC curves (and AUCs) is really what I would want as a method user. One would typically plot TPR versus achieved FDR to get an idea of how well a method is functioning. In general, these are somewhat correlated with AUCs, but TPR versus FDR curves are more typically used in this situation. And, because we would want to see how the error control functions relative to the cutoffs used (i.e., is a $FDR=.05$ cutoff giving a 5% FDR?).

Responses to reviewer comments

We thank the Editor and the two reviewers for their constructive comments. We appreciate the positive comments, such as “the paper reads nicely and is well-organised”. Importantly, we took all concerns raised by the reviewers seriously. In order to address those concerns, we have performed substantial amount of additional computational benchmarking experiments with additional data sets, methods that we compare against, and make following changes:

1. Benchmarked our method with an additional method, ANCOM-BC.
2. Included additional real and simulated data sets to further demonstrate the usability of DCATS
3. Added new function of DCATS that allows a set of reference cell types for normalization and an additional analysis that evaluates the performance of DCATS and other methods in turn of large changes in compositionality.
4. Improve the overall quality of writing by adding more description on the simulation process and the evaluation procedure. We also make our code publicly available in a GitHub repository to ensure our work can be readily reproduced.

All results in the revised manuscript support the high performance and flexibility of DCATS for differential composition analysis. We also confirmed the abstract is under 100 words.

Below, we provide point-by-point responses to the comments by the editor and the reviewers. Their comments are displayed in **blue italics** in this response letter, and we highlight our modifications in **red** in the revised manuscript.

REVIEWER #1

Reviewer #1: The authors present DCATS, a new method for differential abundance (DA) testing of cell type population data derived from single-cell RNA sequencing (scRNA-seq). The method consists of two steps: first, a corrected abundance table is computed trying to fix clustering errors by leveraging a dissimilarity matrix. Then, differential abundance testing with a Beta-Binomial regression model is performed on the corrected abundances. The method achieved good performances on simulated data and was also tested on real scRNA-seq datasets.

Response: We thank the reviewer for the concise summary and insightful comments.

However, the authors' contribution is only marginal, as Beta-Binomial models for abundance data have been proposed and discussed before also in the context of single-cell data (Büttner et al., 2020; Martin et al., 2020).

Furthermore, the model disregards the compositionality of the data and it is not clearly demonstrated that the abundance correction step improves the cell type classification and therefore DA testing.

Also, we are not satisfied with the detail of description and scope of experimental setups, that a)

do not allow to properly judge the method, and b) are insufficient to fully reproduce the experiments.

Response: We agree that using Beta-Binomial models in the context of single-cell data is not novel. Except from using beta-Binomial models, DCATS also includes:

1. The bias correction of misclassification part using a similarity matrix of cell types, as well as two simple but effective methods to estimate the similarity matrix. We believe this part is novel especially in single cell RNA sequencing area;
2. A strategy to estimate the dispersion term for all cell types jointly, through a pooled beta-binomial regression in a pre-step;
3. A function to use reference cell types for normalization, and a way to automatically detect reference cell types.

In fact, new simulation data show that DCATS indeed perform very well in the presence of changes in overall compositionality of the data (see below).

The manuscript has been substantially revised to ensure details of description and scope of experimental setups are clearly described and easily reproduced.

Major comments:

1) Although single-cell abundance data are clearly compositional and the authors acknowledge this at multiple occasions in the manuscript (sum-to-one constraint in M , Dirichlet-Multinomial sampling in the benchmark), they completely miss this property when testing for differential abundance. As shown by (Büttner et al., 2020) in their scCODA manuscript, a Beta-Binomial model suffers from the same false-positive inflation properties as the other tests used in the benchmarking setup (t -tests in speckle, Negative Binomial model in milo, ...). Therefore, it would be important to include also statistical tests that model compositionality into the benchmark. While we know of no compositional DA testing methods that incorporate cell type uncertainty, we assume that the adjusted cell count matrix from DCATS could be used as the input to a compositional method like ANCOM-BC (Lin and Peddada, 2020) to have a fair comparison solely between the DA testing steps. Such a comparison is necessary to ensure that the second part of the method does not suffer from flaws that are masked by the improvements gained by cell count adjustment.

Response: In the revised manuscript, we included ANCOM-BC for comparison and discussed the results in section 2.3.

“By comparing the results of ANCOM-BC using the observed proportion (ancombc) and bias-corrected proportion estimated by DCATS (bcancombc), we found that the bias correction step always increased the sensitivity of ANCOM-BC (Supp. Table S1,3). However, the specificity sometimes decreased, and this led to the overall similar performance of ancombc and

beancombc. While in DCATS, using bias-corrected proportions led to a higher increase in sensitivity and almost no decrease in specificity (Supp. Table S1,3).”

For the systematic assessment of compositional DA, see our response to the point 4 below.

2) The authors fail to show that the clustering bias corrections used in DCATS give an improvement in classification when compared to the cell type composition of a real biological sample. All that is achieved by adjusting the abundance table with the help of a clustering similarity matrix is to smooth the abundance data towards the results produced by another classification method (KNN or SVM). It is necessary to validate with real data whether this step actually reduces the errors in cell type clustering/classification produced by, e.g., Louvain clustering.

Response: We are grateful to the reviewer for the comment. DCATS does not change the clustering/classification results directly. Instead, DCATS estimates a bias corrected abundance based on a similarity matrix. In the revised manuscript, we discuss the performance of DCATS using a real-world data in section 2.2 (page 4 and copied below), where the “ground-truth” cell type annotations are taken from CITE-seq with both transcriptome + protein markers.

Based on the results in section 2.2 we can find that if we know the true similarity matrix, DCATS can provide the estimation of proportion very close to the true proportion (root mean square error (RMSE) from 0.1810 to). But in practice, we could not know the true similarity matrix. DCATS provide simple and fast methods to estimate this similarity matrix, which improve our estimation of proportion by 44% (RMSE from 0.1810 to 0.1023). So it is still valuable to include bias correction in DCAST when doing differential abundance analysis. How to estimate a similarity matrix remains an open area for further exploration.

“We further asked whether the bias correction component in DCATS could correct the misclassification and provides more accurate proportion estimations. We test this using a single cell RNA-seq data set [25] consisting of progressive multiple sclerosis (MS) and relapsing-remitting disease course MS (RRMS). In this dataset, cells from 71 PBMC samples of 62 donors are collected for single-cell RNA sequencing and surface antibody staining. In total, it contains 497,705 single-cell transcriptomic (Tr) and 355,433 surface protein (SP) profiles. The original study used defined cell types based on both Tr and SP profiles [25]. We treat this Tr+SP annotation as the ‘original’ annotation with high reliability, as it is the most well-studied annotation [25].

To test bias correction in our study, we focused on four T-cell subtypes based on the Tr+SP annotated dataset, namely T06, T07, T09, and T10. Then, we identify cell clusters based on transcriptomic data only (Tr-only) and match each Tr-only cluster to its closest Tr+SP cluster. A confusion matrix can be generated between the four clusters defined by Tr+SP and their matching Tr-only clusters. The proportions of four Tr-only clusters can be calculated. Using this ‘empirical’ confusion matrix for bias correction, DCATS can corrected the proportions of four Tr-only clusters very close to the true proportions (root mean square error (RMSE) from 0.1810 to 8.672×10^{-5} , Supp. Fig. S3). In most cases, the true similarity matrix or confusion matrix is

unknown. DCATS provides two simple methods to approximate this similarity matrix, which improves our estimation of proportion by 44% compared to without bias correction (RMSE from 0.1810 to 0.1023, using KNN similarity matrix; Supp. Fig. S3).

Combined with previous simulation results, our findings support the effectiveness of DCATS' bias correction step in differential composition analysis.”

3) It is unclear whether the improved performance of DCATS stems from the correction of misclassifications in the abundance data or from a better DA testing model. Therefore, it is necessary to compare the methods before and after applying the bias correction step from DCATS, wherever applicable, to distinguish whether the bias correction or the Beta-Binomial model are the main sources of improvement.

Response: We already have this comparison in Supplementary Figure S6 and Supplementary Table S1,3. We compared the performance with and without bias correction. We have now further clarified the results in the revised manuscript (section 2.3). In short, DCATS (with or without bias correction) performs better (higher mcc and AUC) than other methods. Based on the improvement in the value, we would like to conclude that the beta-binomial part is the main source of improvement, while the bias correction part can contribute to further improvement.

4) There are some passages in the benchmark description that were not clear to us. First, the description of the theoretical simulation (Section 4.4) misses many important details (e.g., number of replicates, sample size, numerical values) or it is not clear from the description what is meant (What are 1S, 2S and how are 2 and 3 vectors simulated from them? Are those the sample sizes?).

The same is true for Section 4.5 (What is the range of total cell counts per sample? Is the base composition coming from splatter? How big are the differential effects?)

Further, the model comparison study does not disclose the number of replicates. Multiple datasets should be simulated for each sample size and dataset size to avoid biases. It is not clear from the methods section if the benchmark was replicated.

Finally, the description of the simulation with confounders does not disclose which cell types were affected by the confounders (presumably the same as the ones affected by the condition, but this is not clear) and whether the simulations were performed with one confounder (age/gender) each or also with both confounders at the same time. Also, information on effect sizes and value distributions of the confounders is missing.

Also, it is necessary to see a simulation where all DA cell types are increasing, to showcase that DCATS can deal with the spurious effects arising due to the compositionality of the data.

Response: We revised section 4.4 and 4.5 accordingly. We also added a new simulation case with proportion increase in all DA cell types in section 2.3 and discussion in section 3.

“Next, we conduct the second type of simulations to assess the composition effects in task. Namely, when considering the composition problem, it is natural to find that when one cell type increase, the proportion of the rest cell types will decrease. But this kind of proportion decrease is not an actual abundance change.

We simulated 8 cell types with count units [1, 4, 4, 4, 4, 4, 4, 4] in condition 1 and [20, 4, 4, 4, 4, 4, 4, 4] in condition 2. Namely, the first cell type has a 20 times count increase in condition 2, while the rest cell types keep the same. Both conditions contain 3 replicates (see more details in Methods). When the total count remains as 3000 in both condition 1 and condition 2, the observed cell count of the first cell type shows a huge increase in condition 2, while cell counts of the rest cell counts decrease (Supp. Fig. S11). We test the performance of different methods using this dataset. With the estimated dispersion term, this systematic decrease of proportion can be partially modelled by DCATS (DCATS (estPhi emK): 0.762 vs wtoPhi emK: 0.190 in MCC, Fig. 2H, Supp. Table S6). Thus, DCATS achieves higher MCC and F1 compared to other methods.

To directly address the proportion issue, we explore the performance of selecting some confident non-differential cell types as the reference group for normalisation, instead of using all remaining cell types (see more details in Methods). Using some cell types as the reference can further help DCATS distinguish an actual abundance change from the spurious effect caused by other cell types. To increase the stability of normalization as much as possible, we recommend using more than 1 cell type and more than 20% of total cells as the reference. In this simulation, using 3 known unchanged cell types as the reference group, the performance of DCATS further improved (0.762 to 0.948 in MCC, 0.775 to 0.954 in F1) mainly through the decrease of specificity (0.917 to 0.986, Supp. Table S6). Such improvement is also seen when using 3 reference cell types automatically recommended by DCATS (0.762 to 0.889 in MCC, 0.775 to 0.899 in F1).

Meanwhile, ANCOMBC[11] only detected the first cell types as the differential abundance cell type, it was not influenced by this spurious effect, and shows the highest MCC (0.965) and F1 (0.969).”

“Lastly, the differential abundance of cell type naturally regards relative abundance. Hence, choosing the reference (i.e., non-differential cell types) can make an impact. Typically, we assume that the cell type abundance change is relatively small, hence using all remaining cell type is our default setting and demonstrate robust performance in broad scenarios. On the other hand, if the compositional change is substantial, all non-differential cell types will be affected and selecting some cell types as a confident reference can reduce such impact, e.g., by DCATS’s automatic recommendation. Nonetheless, such a large compositional difference is not a typical scenario for DCATS and we suggest users perform the reference cell selection with caution.”

5) In the real data applications, it is dangerous to take the results reported by the original publications as a "ground truth". These results were obtained with inadequate statistical models, which the authors correctly point out (e.g., for the Haber et al. dataset). Therefore, simply obtaining the same results as the initial publication is not a good measurement of method performance, and only points out that the method may suffer from the same flaws.

Because of this, we think that a judgment of methods based on their similarity with each other is more effective than comparing to one (likely inaccurate) result.

Response: We agree that the conclusions given by the original publication cannot be treated as ground truth in the Haber et al. Dataset. The goal of the evaluation is to show that DCATS can automatically identify sensible differentially abundant cell types, and therefore we treat the annotation in the original paper as a silver standard in which the results of DCATS and other methods can be compared against. DCATS is the method that provides the most similar results to the major votes.

In the paper, we benchmark different methods using two datasets coming from the publications. Another dataset, Kang2017, should not have composition differences between two conditions according to the original paper. This conclusion is based on the biological knowledge that IFN- β will not affect cell type composition between two conditions. To further validate the performance of DCATS, we apply DCATS in the Angelidis2019 dataset. Details are discussed in section 2.5.

“We then assessed the sensitivity of DCATS using a dataset (Angelidis2019) containing whole lungs from 3-month-old mice (n=8) and 24-month-old mice (n=7) with single-cell transcriptomic and mass spectrometry-driven proteomic data [29]. In this dataset, authors discovered one cluster of cells had a high expression level of S and G2M cell-cycle marker genes. They are mainly proliferating cells, thus, should show higher abundance in young mice. As these cells belong to T cells, type-2 pneumocytes, and alveolar macrophages, they should have higher proportions in young mice. Using single-cell transcriptome information, DCATS enabled the detection of type-2 pneumocytes, and alveolar macrophages, and a subtype of T cells, CD4+ T cells, as differentially abundant cell types. (Supp. Table S8) In addition, the authors found a significantly altered ratio of club to ciliated cells in aged mice, which was experimentally validated [29]. Using club cells as the reference group, DCATS enabled detection of this ratio change ($p = 0.0053$, Supp. Table S8).

The third dataset consists of 53,193 epithelial cells from mice’s small intestine and organoids [27]. It has 4 control samples, 2 samples coming from two days after Salmonella infection, 2 samples coming from three days after H.polygyrus infection, and 2 samples coming from ten days after H.polygyrus infection. Eight cell types were defined in the original study and then were compared between controls and each stimulation group to identify cell types with differential composition abundance through a Poisson regression and Wald test [27]. Presumably because the sample variability is not well considered in the Poisson regression, the authors used $FDR < 1 \times 10^{-5}$ as the cutoff of differential abundant cell types between control and simulations. We do not have the ground truth about which cell types are differentially abundant, but for the purpose of evaluating the accuracy of DCATS and other methods, we used the findings in the original study of this dataset as the ‘silver standard’. When re-analysing this data set, we first noticed that Fisher’s exact test captured all the reported differential abundant cell types, but also reports lots of differential abundant cell types that are regarded as non-differentially abundant cell types by other methods in each of the comparison pairs (with $p < 0.01$). Similarly, scDC and milo also returns ‘false positives’ in each comparing condition pair but occasionally missed reported positives. Surprisingly, speckle shows limited power with missing almost all hits. In contrast, DCATS shows a good balance between sensitivity and specificity; it does not return any ‘false positive’ (even at a lenient significance level 0.1) and captures most of the reported differential abundant cell types (Fig. 3B-D, Supp. Table S10).

Furthermore, if we treat the results given in the paper as results of another method, we can use the major decisions as highly possible conclusions. In this case, the performance of each method is evaluated by their similarity with other methods. We can find that DCATS provides a consistent conclusion with the majority decisions of each cell type in different comparisons, except for the significant proportion change of Goblet between control and Hpoly day 10 conditions. “

6) When showing performance indicators obtained on multiple replicates like F1 score or MCC, some sort of confidence interval or standard deviation should be reported. These values are missing in the entire manuscript.

Due to the small sample sizes, confidence intervals will be relatively large. While this is always true for testing on datasets with low replicates, it still has to be discussed.

Response: We used bootstrap to sample the simulation results 50 times and calculate the confidence interval of MCC, F1 score, AUC, and PRAUC. The results is in the supplementary table 2 and 4.

Minor comments:

1) The notation between Sections 4.1 and 4.3 is inconsistent. x is used for both cell abundances and covariates, while μ is used for both the relative abundances and the Beta-Binomial mean. Please make sure that the notation is easily understandable and all values are defined.

2) Some heatmaps (Fig. 2D and S4) are color-coded by the method rank instead of the numerical value of the metric, which can be very misleading. If possible, the values of the metric should be used as a color scale.

3) On page 9, there is a typo: Instead of "Language multiplier", it should read "Lagrange multiplier".

Response: We appreciate the comment by the reviewer. We revised corresponding sections accordingly.

References

- 1. Büttner, M., Ostner, J., Müller, C. L., Theis, F. J., and Schubert, B. (2020). scCODA: A Bayesian model for compositional single-cell data analysis. bioRxiv, 2020.12.14.422688. doi:10.1101/2020.12.14.422688.*
- 2. Lin, H., and Peddada, S. D. (2020). Analysis of compositions of microbiomes with bias correction. Nat. Commun. 11, 3514.*
- 3. Martin, B. D., Witten, D., and Willis, A. D. (2020). MODELING MICROBIAL ABUNDANCES AND DYSBIOSIS WITH BETA-BINOMIAL REGRESSION. Ann. Appl. Stat. 14, 94-115.*

REVIEWER #2

Reviewer #2: Lin and colleagues present an interesting new tool called DCATS for differential abundance analysis that under the hood uses beta-binomial modelling as well as some bias correction. Overall, the paper reads nicely and is well-organised. I have a small number of comments that I think would be worth clarifying or addressing.

Response: We are grateful to the reviewer for the concise summary and the positive comments.

1. There is mention in multiple places near the Introduction about 'complex experimental design', 'between two conditions'. But, existing frameworks handle all this and have done so for years. So, I find this aspect of motivation somewhat strange (related comment below).

Response: We agree that complex is not the most suitable word. We change it to flexible since DCATS can be easily adopted in different experimental designs.

2. Why does Fisher's exact test get such a big mention? It's quite clear that this is not even a reasonable baseline, because it is not capable of handling replicates. It would've seemed more logical (especially for a statistics paper) to have text at the beginning to say that Fisher's exact is not appropriate; end of story. I would suggest to make a clear statement about the appropriateness of FET in this context.

Response: In the revised manuscript, we point out that Fisher's exact test is not appropriate for differential abundance analysis.

“Fisher exact test cannot fully utilize the biological replicates (if any) to account for biological variability, which makes it not appropriate when more than one biological replicates exist.”

3. Later in the paper, propeller and diffcyt are tested against DCATS. So, I find it surprising that these approaches are not mentioned in the introduction. Usually in an introduction, the entire space of relevant methods would be spelled out to the reader, not just a subset. I would suggest to expand the Introduction to have complete coverage of the methods in this space.

Response: We appreciate the valuable suggestion by the reviewer. In the revised manuscript, we add more content about the method in this space.

“Some statistical tools are also developed specialized for scRNA-seq data. scCODA [7] assumes cell counts of different cell types follows hierarchical Dirichlet-Multinomial distribution, which allows scCODA to model all cell types together. MiloR [8] evaluates differential abundance on smaller clusters in KNN graph, which are called 'neighbourhoods'. It assumes that cell counts follow a negative binomial distribution and uses representative cells instead of all cells to improve program efficiency. Daseq [9] also makes use of KNN graph and it calculates

multiscale differential abundance scores by counting the numbers of cells coming from different biological states while varying k . This multiscale differential abundance score is what DAsseq uses to infer cell states that have differential abundance. propeller [10] uses an empirical Bayes framework to enable information sharing between different samples and chooses different statistical methods (t-test or ANOVA) based on the number of conditions. Other tools are developed to handle different challenges for different data types. ANCOM-BC [11] uses a linear model with sample-specific offset terms to handle the estimation of different sampling fractions when analyzing microbiome data. diffcyt[12] uses functions from the R package edgeR [13] to test differential abundance of cell types. It uses an overdispersed Poisson model with empirical Bayes methods for information sharing between different cell types. However, the code was specifically designed for flow cytometry data, not single cell RNA-seq data.”

4. From a brief look, the R package looks well designed and documented, but I would suggest also that the authors submit their package to Bioconductor. This typically results in an increase in the standard of packages and has some extra requirements in terms of interoperability with other packages, etc. It also has the benefit of additional exposure.

Response: We submitted DCATS to Bioconductor on Nov 22. We are actively communicating with Bioconductor to make sure DCATS will be accepted.

5. For me, in a methods paper, the result is the method. Thus, in the first section of the Results, I would suggest to give more mathematical details of the method, including the parameterisation of the BB model and a few more details of the error correction / similarity matrix calculation (in the second section of the Results). In addition, the papers suggests that DCATS has 2 novel features, but I'm not sure if either are really "novel" (I also think the authors don't need to "sell" the novelty; they propose a clever and reasonable approach and show that it works well; I would always suggest to focus on the results, not the salesmanship). For example, BB modelling has been used successfully in DNA methylation analysis for a long time:

<https://bmcbioinformatics.biomedcentral.com/articles/10.1186/1471-2105-15-215>

Perhaps also worth mentioning is their bias correction has similarity to that done in other 'differential abundance' contexts, such as microbiome analysis:

<https://www.nature.com/articles/s41467-020-17041-7>

(Also, there is a lot of literature on more-or-less the same topic in microbiome analysis that is not mentioned in the Introduction. Also, there is a lot of literature on 'compositional' analysis that is not well-covered in the Introduction.)

Also, given that "composition" is mentioned several times in the paper, it would make sense to compare their method against a compositional data analysis method, such as ANCOM-BC.

Response: We agree that using beta-binomial is not a new thing in bioinformatics tools development in sequencing data. We revised manuscript to avoid the misused of word ‘novel’. Still, we believe the bias correction for misclassification is novel, especially in single cell transcriptomics data analysis. and its effective inference with EM algorithm. We appreciate the reviewer let us know about ANCOM-BC. We also add the comparison of ANCOM-BC for benchmark. But it is worth noting that the bias correction of ANCOM-BC is not for misclassification, instead, it is for correcting different sampling fraction.

6. I think we need more details about how other (competing) methods were run. The statement "[Method X] fails to control influence of confounding covariates although we included related information when performing the analysis" is very surprising. Several of these methods are based on linear modelling frameworks such that they can adjust for covariates. And they magically all fail to do this? I find this suspicious. At the very minimum, the authors should make all the code available that they used to run these methods. But also some detective work would be nice to understand why all other methods fail. It simply cannot be that diffcyt, speckle, milo are completely unable to account for covariates (Figure 2E).

Response: We put the link of github repository in the code availability part. Except for DCATS, scDC, diffcyt, and milo also use linear modelling framework. But scDC only allows to input one vector indicating the condition for comparison. diffcyt uses edgeR in its tutorial for differential abundance analysis, but it starts from the clustering based on flow cytometry data. It is challenging to add additional covariates information for testing. Milo enables to include the information of confounding covariates, but it cannot provide the differential abundance analysis regarding to different covariates simultaneously. We updates the results given by milo in Figure 2 and section 2.3.

“Even though many methods are based on a linear framework, DCATS, milo, and ANCOMBC are the only three methods designed to support additional covariates testing. ANCOMBC only supports testing for discrete covariates. scDC, as a GLM based method, is in principle able to support covariates but the interface is not implemented. Milo and ANCOMBC, which are all based on GLM models, fail to control influence of confounding covariates although we included related information when performing the analysis. diffcyt, which uses edgeR [13] for differential abundance analysis on flow cytometry data is also challenging to incorporate different covariates due to its highly specific analysis pipeline designed for flow cytometry data.”

7. Scenarios: One thing that is noticeable from the simulations is that the ground truth changes introduced are offsetting. That is, some subpopulations increase in relative abundance, while others decrease. This is convenient, because it helps make the normalisation easier. But, what about a situation where one cell type expands in magnitude? (this is actually a common situation in immunological contexts). Here (i.e., one subpopulation expanding in magnitude), the normalisation is more challenging, because in a relative sense, all subpopulations now change

(unless it's normalised out). Would DCATS handle such a situation?

Response: We agree it is important to know the performance of DCATS in this kind of scenario. We add the corresponding results in the section 2.3 and discussed in the discussion part.

“Next, we conduct the second type of simulations to assess the composition effects in task. Namely, when considering the composition problem, it is natural to find that when one cell type increase, the proportion of the rest cell types will decrease. But this kind of proportion decrease is not an actual abundance change.

We simulated 8 cell types with count units [1, 4, 4, 4, 4, 4, 4, 4] in condition 1 and [20, 4, 4, 4, 4, 4, 4, 4] in condition 2. Namely, the first cell type has a 20 times count increase in condition 2, while the rest cell types keep the same. Both conditions contain 3 replicates (see more details in Methods). When the total count remains as 3000 in both condition 1 and condition 2, the observed cell count of the first cell type shows a huge increase in condition 2, while cell counts of the rest cell counts decrease (Supp. Fig. S11). We test the performance of different methods using this dataset. With the estimated dispersion term, this systematic decrease of proportion can be partially modelled by DCATS (DCATS (estPhi emK): 0.762 vs wtoPhi emK: 0.190 in MCC, Fig. 2H, Supp. Table S6). Thus, DCATS achieves higher MCC and F1 compared to other methods.

To directly address the proportion issue, we explore the performance of selecting some confident non-differential cell types as the reference group for normalisation, instead of using all remaining cell types (see more details in Methods). Using some cell types as the reference can further help DCATS distinguish an actual abundance change from the spurious effect caused by other cell types. To increase the stability of normalization as much as possible, we recommend using more than 1 cell type and more than 20% of total cells as the reference. In this simulation, using 3 known unchanged cell types as the reference group, the performance of DCATS further improved (0.762 to 0.948 in MCC, 0.775 to 0.954 in F1) mainly through the decrease of specificity (0.917 to 0.986, Supp. Table S6). Such improvement is also seen when using 3 reference cell types automatically recommended by DCATS (0.762 to 0.889 in MCC, 0.775 to 0.899 in F1).

Meanwhile, ANCOMBC[11] only detected the first cell types as the differential abundance cell type, it was not influenced by this spurious effect, and shows the highest MCC (0.965) and F1 (0.969).”

“Lastly, the differential abundance of cell type naturally regards relative abundance. Hence, choosing the reference (i.e., non-differential cell types) can make an impact. Typically, we assume that the cell type abundance change is relatively small, hence using all remaining cell type is our default setting and demonstrate robust performance in broad scenarios. On the other hand, if the compositional change is substantial, all non-differential cell types will be affected and selecting some cell types as a confident reference can reduce such impact, e.g., by DCATS’s automatic recommendation. Nonetheless, such a large compositional difference is not a typical scenario for DCATS and we suggest users perform the reference cell selection with caution.”

8. In Figure 2C, the lines are impossible to see unless you go to 300% magnification on a large screen. I would also suggest to colour the x with the corresponding method. I also wonder whether ROC curves (and AUCs) is really what I would want as a method user. One would typically plot TPR versus achieved FDR to get an idea of how well a method is functioning. In general, these are somewhat correlated with AUCs, but TPR versus FDR curves are more typically used in this situation. And, because we would want to see how the error control functions relative to the cutoffs used (i.e., is a $FDR=.05$ cutoff giving a 5% FDR?).

Response: We revised this plot accordingly. The TPR vs FDR plot is the Supplementary Figure S5.

Second round of review

Reviewer 1

We thank the authors for revising their manuscript and making substantial improvements to the completeness of the method's performance assessment, as well as addressing our previous concerns. In its present form, we consider the manuscript as a valuable contribution to the field but have a couple of minor questions and suggestions.

Comments

- 1) In the introduction, the t-test is listed as a representative statistical test for compositional data, although clearly the presumptions of the t-test are violated. Please give a more appropriate example.
- 2) The paragraph discussing Fisher's exact test in the introduction seems to be detached from the rest of the introduction. We suggested either integrating the discussion better or removing this paragraph.
- 3) The description of the model in the results section is rather incomplete. We would recommend writing down the full hierarchical model including the parametrization of \bar{p}_{ij} so that the reader is not forced to puzzle together the model from the partial description and Figure 1.
- 4) We thank the authors for acknowledging that taking the results from Haber et al. as a comparison baseline is not a good idea. However, all the methods used for comparison do not account for compositionality in the data, and could therefore systematically produce the same errors. We argue that in the Salmonella case, the decreases in Stem, TA, and TAE cells are due to the stark relative increase of Enterocytes and should not be detected. This should be discussed in the paper, e.g., by additional comparisons with the results of ANCOM-BC and DCATS with normalization. In addition, such a comparison would also make for a great showcase of the normalization wrt. a reference on real data.
- 5) We appreciate the authors' effort to compare their method to a true compositional method in ANCOM-BC, and also include a simulation scenario that shows the problem of most methods with compositional effects. Unfortunately, the additions to Sections 2.3, 4.4, and 4.8 discussing this simulation are not living up to the rest of the manuscript in terms of quality of language and are thus not as easy to follow (e.g., "Namely, when considering the composition problem, it is natural to find that when one cell type increase, the proportion of the rest cell types will decrease."). We would appreciate it if these sections would be improved to reach the excellent stylistic tone of the rest of the manuscript.

6) The sentence "Even though many methods are based on a linear framework, DCATS, milo, and ANCOMBC are the only three methods designed to support additional covariates" (Page 7) leaves the impression that only the mentioned methods in general here can handle covariates, although, in fact, the authors talk specifically about the once explicitly tested in the benchmark. We suggest clarifying this here.

We acknowledge the authors' method of normalizing the data with respect to a reference as a good way to approximately deal with compositionality effects, which is confirmed by the simulations. It would be great if the authors could improve the description of how to perform the normalization in Section 4.4 with a mathematical formula to make the approach easier to understand, especially when using multiple cell types as the reference.

7) The caption for Figure 2C mentions a dashed black line at FDR=0.05, which is missing in the plot. (although we're not quite sure how this line fits into the plot anyways) Does this sentence belong to Figure S5 instead (it's missing there)?

8) Please ensure consistent usage of present and past tense throughout the manuscript (e.g., in Sections 4.5 and 4.6).

9) Please also check for typos again. For example in Section 4.6: "The cluster annotation of each cell is treated as input for the downstream analysis." should read "[...] downstream analysis."

Reviewer 2

Overall, the authors have done an excellent job of addressing my and the other reviewer's concern. This is one of the best revisions I've seen, in terms of clarity and completeness of the response.

There is one aspect of the response that I would like to dig a little deeper into. There was a comment about additional covariates not being able to be adjusted for, in both miloR and diffcyt. For the latter (diffcyt), I understand that the implementation is linked with clustering analysis of cytometry data, but still, the (corrected) counts could be used directly with edgeR. That is, the statistical engine is edgeR and could be called directly. So, I don't agree with the statement that "it is challenging to add additional covariate information for testing". It is not difficult.

For miloR, the code used is:

https://github.com/linxy29/DCATS_analysis/blob/master/test_milo.Rmd#L164-L169

The authors comment "[Milo] fails to control influence of confounding covariates although we included related information" .. the code shows all kinds of models that the user is testing, but it's not clear the testing is done correctly, because no contrasts are specified. Again here, the engine underneath is edgeR (but could easily be any other) and thus the counts could be extracted and the testing done in whatever way is desired. The docs (for miloR) make this clear: "This function wraps up several steps of differential abundance testing using the edgeR functions. These could be performed separately for users who want to exercise more control over their DA testing."

Also, propeller is not mentioned in this section and should also "support additional covariates testing".

Responses to reviewer comments

We thank the Editor and the two reviewers for their constructive comments. We appreciate the positive comments, such as “we consider the manuscript as a valuable contribution to the field”. We have carefully reviewed the comments and suggestions from the reviewers. We have provided additional explanations and made the necessary revisions to improve the clarity and readability of the paper. We also provided extra experiments to support the high flexibility of DCATS.

Below, we provide point-by-point responses to the comments by the editor and the reviewers. Their comments are displayed in *blue italics* in this response letter, and we highlight our modifications in **red** in the revised manuscript.

Reviewer #1

Reviewer #1: We thank the authors for revising their manuscript and making substantial improvements to the completeness of the method's performance assessment, as well as addressing our previous concerns. In its present form, we consider the manuscript as a valuable contribution to the field but have a couple of minor questions and suggestions.

1) In the introduction, the t-test is listed as a representative statistical test for compositional data, although clearly the presumptions of the t-test are violated. Please give a more appropriate example.

Response: We thank the reviewer for your careful reading and attention to detail. T-test has been used in the literature, such as Kaufmann et al [1], but we agree this may not be a typical choice of statistical test in general cell composition analysis. Therefore, we removed this example from the text.

2) The paragraph discussing Fisher's exact test in the introduction seems to be detached from the rest of the introduction. We suggested either integrating the discussion better or removing this paragraph.

Response: We agree that the paragraph discussing Fisher's exact test in the introduction seems to be detached from the rest of the introduction. After careful consideration, we decided to remove this paragraph from the manuscript to improve the flow and coherence of the introduction.

3) The description of the model in the results section is rather incomplete. We would recommend writing down the full hierarchical model including the parametrization of \bar{p}_{ij} so that the reader is not forced to puzzle together the model from the partial description and Figure 1.

Response: We agree that a full hierarchical model description would enhance the clarity of our results and facilitate readers' understanding. Therefore, we added a complete hierarchical model in the result section and modified Figure 1 to improve the consistency of notations. The following text has been added in the Result section (page 2-3):

“We assume the corrected cell counts $Z_{s,j}$ for cell type j given sample s follow beta-binomial distribution. We then describe $Z_{s,j}$ with a beta-binomial generalised linear model (GLM) with a logit link function for each covariate i , as follows,

$$\begin{aligned} z_{s,j} &\sim \text{Binom}(n_s, p_{s,j}) \\ p_{s,j} &\sim \text{Beta}(\alpha_{s,j}, \beta_{s,j}) \end{aligned}$$

where we reparameterized with the mean $p_{s,j}^-$ and over-dispersion term ϕ_j as

$$\begin{aligned}\alpha_{s,j} &= p_{s,j}^- (1/\phi_j - 1) \\ \beta_{s,j} &= (1 - p_{s,j}^-) (1/\phi_j - 1)\end{aligned}$$

As described below, the mean parameter $p_{s,j}^-$ is regressed to a set of covariates under different hypotheses. $p_{s,j}^-$ can be parameterized as

$$\text{logit}(p_{s,j}^-) = w_0 + w^T c_s$$

where c_s are the covariate vectors for sample s , and w are the weights of covariates”

4) We thank the authors for acknowledging that taking the results from Haber et al. as a comparison baseline is not a good idea. However, all the methods used for comparison do not account for compositionality in the data, and could therefore systematically produce the same errors. We argue that in the Salmonella case, the decreases in Stem, TA, and TAE cells are due to the stark relative increase of Enterocytes and should not be detected. This should be discussed in the paper, e.g., by additional comparisons with the results of ANCOM-BC and DCATS with normalization. In addition, such a comparison would also make for a great showcase of the normalization wrt. a reference on real data.

Response: We agree. In response, we performed an additional comparison with the results of ANCOM-BC and DCATS with normalisation using the reference group. The results show that enterocyte is the only cell type with differential abundance. This new result is described in the Result section (page 10-11):

“When comparing the control groups and *Salmonella* infection group, we observed a significant increase in the proportion of enterocytes, which causes a proportion decrease in the rest of the cell types. Using the reference group detected by DCATS as the normalization term, it was able to overcome the influence of enterocytes. The results show that enterocyte is the only cell type with differential abundance (Fig. 3B-D, Supp. Table S9-10, Supp. Fig. 14).”

5) We appreciate the authors' effort to compare their method to a true compositional method in ANCOM-BC, and also include a simulation scenario that shows the problem of most methods with compositional effects. Unfortunately, the additions to Sections 2.3, 4.4, and 4.8 discussing this simulation are not living up to the rest of the manuscript in terms of quality of language and are thus not as easy to follow (e.g., “Namely, when considering the composition problem, it is natural to find that when one cell type increase, the proportion of the rest cell types will decrease.”). We would appreciate it if these sections would be improved to reach the excellent stylistic tone of the rest of the manuscript.

Response: We modified the Results section (including Section 2.3) and the Method section (including Section 4.4, 4.8) to improve clarity.

6) The sentence “Even though many methods are based on a linear framework, DCATS, milo, and ANCOMBC are the only three methods designed to support additional covariates” (Page 7) leaves the impression that only the mentioned methods in general here can handle covariates, although, in fact, the authors talk specifically about the once explicitly tested in the benchmark. We suggest clarifying this here.

Response: We revised the sentence on page 7 to clarify the scope of the benchmarking analysis: “Even though most of these seven methods are based on a linear framework, DCATS, milo, and

ANCOMBC are the only three methods designed to support additional covariates testing.” (see page 8 of the revised manuscript)

We acknowledge the authors’ method of normalizing the data with respect to a reference as a good way to approximately deal with compositionality effects, which is confirmed by the simulations. It would be great if the authors could improve the description of how to perform the normalization in Section 4.4 with a mathematical formula to make the approach easier to understand, especially when using multiple cell types as the reference.

We also updated the mathematical description in the Method section (page 16) to improve clarity:

“When using the total cell number as the normalization term, we have

$$\text{logit}(E(\frac{Z_{s,j}}{n_s})) = w_0 + w^T c_s$$

where values in w^T could be zero based on different models. When using the reference group as the normalization term, we have

$$\text{logit}(E(\frac{Z_{s,j}}{\sum_{r \in R} Z_{s,r}})) = w_0 + w^T c_s$$

where R denotes the reference group.”

7) The caption for Figure 2C mentions a dashed black line at FDR=0.05, which is missing in the plot. (although we’re not quite sure how this line fits into the plot anyways) Does this sentence belong to Figure S5 instead (it’s missing there)?

Response: That sentence referred to Figure S5. We have now moved it to the caption of Figure S5.

8) Please ensure consistent usage of present and past tense throughout the manuscript (e.g., in Sections 4.5 and 4.6).

Response: We went through the paper again to ensure consistent usage of present and past tense throughout the manuscript.

9) Please also check for typos again. For example in Section 4.6: “The cluster annotation of each cell is treated as input for the downstrain analysis.” should read “[...] downstream analysis.”

Response: We corrected this error and went through the paper again to avoid the existence of other typos.

[1] Kaufmann, Max, et al. "Identifying CNS-colonizing T cells as potential therapeutic targets to prevent progression of multiple sclerosis." *Med* 2.3 (2021): 296-312.

Reviewer #2

Reviewer #2: Overall, the authors have done an excellent job of addressing my and the other reviewer’s concern. This is one of the best revisions I’ve seen, in terms of clarity and completeness of the response.

There is one aspect of the response that I would like to dig a little deeper into. There was a comment about additional covariates not being able to be adjusted for, in both miloR and diffcyt. For the latter (diffcyt), I understand that the implementation is linked with clustering analysis of cytometry data, but still, the (corrected) counts could be used directly with edgeR. That is, the statistical engine is edgeR

and could be called directly. So, I don't agree with the statement that "it is challenging to add additional covariate information for testing". It is not difficult.

For miloR, the code used is:

https://github.com/linxy29/DCATS_analysis/blob/master/test_milo.Rmd#L164-L169

The authors comment "[Milo] fails to control influence of confounding covariates although we included related information" .. the code shows all kinds of models that the user is testing, but it's not clear the testing is done correctly, because no contrasts are specified. Again here, the engine underneath is edgeR (but could easily be any other) and thus the counts could be extracted and the testing done in whatever way is desired. The docs (for miloR) make this clear: "This function wraps up several steps of differential abundance testing using the edgeR functions. These could be performed separately for users who want to exercise more control over their DA testing."

Also, propeller is not mentioned in this section and should also "support additional covariates testing".

Response: We apologized for the confusion brought by the names of the Rmd file. The code used to apply Milo on the simulation data is in the 'functionsV2.r' file (https://github.com/linxy29/DCATS_analysis/blob/master/functionsV2.r#:~:text=design%20%3D%20~%20age%20%2B%20gender%20%2B%20condition). Files started with 'test' are used to explore the usage of different tools. They are irrelevant to the experiments used in the manuscript. To avoid future confusion, we deleted them in this GitHub repository.

diffcyt uses edgeR for differential abundance testing. edgeR allows the usage of confounding covariates. However, the input of diffcyt is cytometry data instead of single-cell gene expression. To run the whole analysis process of diffcyt, we first generated an input object for diffcyt using random numbers, then replaced information related to the group and sample with actual value. diffcyt supports the usage of multiple covariates, but the overall process of running diffcyt from the beginning is complicated. propeller only allows one vector of input as the group information, so we were not able to add covariates information. We also added this discussion in the manuscript (page 8):

"propeller in the speckle package [10] only accepts input of one vector containing condition information."

Phipson, Belinda, et al. "propeller: testing for differences in cell type proportions in single cell data." *Bioinformatics* 38.20 (2022): 4720-4726.

Third round of review

Reviewer 2

The authors have addressed my concerns.